# The macronutrient and micronutrient (iron and manganese) content of icebergs

Jana Krause[1], Dustin Carroll[2], Juan Höfer[3,4], Jeremy Donaire[5,6], Eric P. Achterberg[1], Emilio Alarcón[4], Te Liu[1], Lorenz Meire[7,8], Kechen Zhu[9], Mark J. Hopwood[9*]

[1] GEOMAR Helmholtz Centre for Ocean Research Kiel, Kiel, Germany
[2] Moss Landing Marine Laboratories, San José State University, Moss Landing, California, USA
[3] Escuela de Ciencias del Mar, Pontificia Universidad Católica de Valparaíso, Valparaíso, Chile
[4] Centro FONDAP de Investigación en Dinámica de Ecosistemas Marinos de Altas Latitudes (IDEAL), Valdivia, Chile
[5] Facultad de Ingeniería, Universidad Andrés Bello, Viña del Mar, Chile
[6] Faculty of Sciences and Bioengineering Sciences, Vrije Universiteit Brussel, Brussels, Belgium
[7] Department of Estuarine and Delta Systems, Royal Netherlands Institute for Sea Research, Yerseke, The Netherlands
[8] Greenland Climate Research Centre, Greenland Institute of Natural Resources, Nuuk, Greenland
[9] Department of Ocean Science and Engineering, Southern University of Science and Technology, Shenzhen, China

*Correspondence to*: Mark J. Hopwood (Mark@sustech.edu.cn)

**Abstract.** Ice calved from the Antarctic and Greenland Ice Sheets or tidewater glaciers ultimately melts in the ocean contributing to sea-level rise and potentially affecting marine biogeochemistry. Icebergs have been described as ocean micronutrient fertilizing agents, and biological hotspots due to their potential roles as platforms for marine mammals and birds. Icebergs may be especially important fertilizing agents in the Southern Ocean, where availability of the micronutrients iron and manganese extensively limits marine primary production. Whilst icebergs have long been described as a source of iron to the ocean, their nutrient load is poorly constrained and it is unclear if there are regional differences. Here we show that 589 ice fragments collected from calved ice in contrasting regions spanning the Antarctic Peninsula, Greenland, and smaller tidewater systems in Svalbard, Patagonia and Iceland have similar (micro)nutrient concentrations with limited or no significant differences between regions. Icebergs are a minor or negligible source of macronutrients to the ocean with low concentrations of $NO_x^-$ ($NO_3^- + NO_2^-$, median 0.51 µM), $PO_4^{3-}$ (median 0.04 µM), and dissolved Si (dSi, median 0.02 µM). In contrast, icebergs deliver elevated concentrations of dissolved Fe (dFe, median 12 nM) and Mn (dMn, median 2.6 nM). Sediment load for Antarctic ice (median 9 mg $L^{-1}$, n=144) was low compared to prior reported values for the Arctic (up to 200 g $L^{-1}$). Whilst total dissolvable Fe and Mn retained a strong relationship with sediment load

(both $R^2 = 0.43$, p<0.001), weaker relationships were observed for dFe ($R^2 = 0.30$, p<0.001), dMn ($R^2 = 0.20$, p<0.001) and dSi ($R^2 = 0.29$, p<0.001). A tight correlation between total dissolvable Fe and Mn ($R^2 = 0.95$, p<0.001) and a total dissolvable Mn:Fe ratio of 0.024 suggested a lithogenic origin for the majority of sediment present in ice. Dissolved Mn was however present at higher dMn:dFe ratios, with meltwater fluxes roughly equivalent to 30% of the corresponding dFe flux. Our results demonstrated that the nutrient concentrations measured in calved icebergs are consistent with an atmospheric source of $NO_x^-$ and $PO_4^{3-}$. Conversely, high Fe and Mn, and occasionally high dSi concentrations, are associated with englacial sediment, which experiences limited biogeochemical processing prior to release into the ocean.

**1 Introduction**

At the interface between the cryosphere and ocean, icebergs are both physical and chemical agents via which ice-ocean interactions affect marine biogeochemical cycles (Enderlin et al., 2016; Helly et al., 2011; Smith Jr. et al., 2007). Icebergs are often described as fertilizing agents, especially in the context of the Southern Ocean (Schwarz and Schodlok, 2009; Smith Jr. et al., 2007; Vernet et al., 2011). However, the fertilizing effect of icebergs is likely regionally dependent due to changes in the identity of the (micro)nutrients limiting marine primary production, and perhaps also due to regional changes in the nutrient load of icebergs. In the Southern Ocean, iron (Fe) is thought to be the main nutrient limiting phytoplankton growth throughout much of the growth season and so changes to regional Fe supply can have ecosystem effects (Martin et al., 1990a, b; Moore et al., 2013). A critical research challenge is therefore to constrain Fe sources and sinks in the Southern Ocean and to assess their climatic sensitivity (Martin, 1990; Wadley et al., 2014). Icebergs are one such Fe source to pelagic ecosystems (Raiswell, 2011; Raiswell et al., 2008; Shaw et al., 2011). Icebergs have long been described as an important Fe source via delivery of both englacial sediment and the dissolved components of ice melt (Hart, 1934; Lin et al., 2011; Raiswell et al., 2008). Positive chlorophyll anomalies following iceberg passage in the Southern Ocean during the growth season have been detected by satellite-derived chlorophyll measurements and these are usually attributed to Fe-fertilization (Schwarz and Schodlok, 2009; Wu and Hou, 2017). However, Fe may not be the only micronutrient to limit marine primary production around

Antarctica. Recent work has suggested that low dissolved manganese (Mn) concentrations are a further co-limiting factor for phytoplankton growth in parts of the Southern Ocean (Browning et al., 2021; Hawco et al., 2022; Latour et al., 2021). As Fe and Mn share similar sources, icebergs might also be an equally important source term for the polar marine Mn cycle (Forsch et al., 2021).

In contrast to Antarctica, Fe-limitation of marine phytoplankton growth in the Arctic is a less prominent feature. Fe-limitation is sparsely reported in the Arctic (Taylor et al., 2013) and largely confined to offshore areas of the high-latitude North Atlantic away from typical iceberg trajectories (Nielsdottir et al., 2009; Ryan-Keogh et al., 2013). Phytoplankton growth within regions around Greenland affected by icebergs is more often limited by nitrate availability (Randelhoff et al., 2020, Krisch et al., 2020). With icebergs thought to supply only limited concentrations of nitrate and phosphate to the ocean, a direct iceberg fertilization effect is not expected in nitrate-limited marine regions (Shulenberger, 1983). The macronutrient content of icebergs is however poorly constrained, especially for components other than Fe. Although subject to large uncertainties, icebergs could be a modest source of silica to the marine environment (Hawkings et al., 2017; Meire et al., 2016) which might have ecological effects. Whilst high macronutrient concentrations are found throughout the Southern Ocean, dissolved silica (dSi) availability often limits diatom growth in the Arctic due to its depletion prior to nitrate (Krause et al., 2018, 2019).

In order to understand how iceberg-derived fluxes of (micro)nutrients may change regionally with climate change and glacier retreat inland, it is necessary to understand the origin and fate of nutrients within calved icebergs at sea. The ultimate origin of nutrients in icebergs could be argued to be atmospheric (Fischer et al., 2015; Hansson, 1994). Inland precipitation and aerosol deposition on ice surfaces will exert a large influence on the nutrient content of bulk ice which is ultimately calved into the ocean as icebergs (Vernet et al., 2011). However, processes beyond the ice-atmosphere interface may also affect the nutrient content of ice. Furthermore, internal cycling may also critically redistribute (micro)nutrients and affect the relative abundance of elements in both dissolved (<0.2 µm) and particulate (>0.2 µm) phases. Landslides onto ice surfaces, and the movement of basal ice over bedrock or subglacial sediments create layers of ice visibly enriched in sediment (Alley et al., 1997; Knight, 1997; Mugford and

Dowdeswell, 2010). Some fraction of the labile phases in these sediments; particularly for the elements Fe, Mn and silica, which are present at high abundances; may ultimately be transformed into bioaccessible nutrients in the ocean (Forsch et al., 2021; Hawkings et al., 2017; Raiswell, 2011). How sediment is gained and lost from ice before, during and after iceberg calving therefore might exert some influence on measured (micro)nutrient concentrations in melting icebergs at sea (Hopwood et al., 2019).

On exposed ice surfaces during the growth season, cryoconite formation and the growth of algae are notable features which will act to re-distribute nutrients between inorganic and organic pools and to amplify heterogeneity in the distribution of nutrients within ice (Cook et al., 2015; Rozwalak et al., 2022; Stibal et al., 2017). These processes will occur alongside, and likely interact with, other photochemical reactions (Kim et al., 2010; Kim et al., 2024). Whilst iceberg calving may temporarily disturb features present on ice surfaces, and the rolling of smaller icebergs will regularly interrupt cryoconite growth on calved ice surfaces, long-lived icebergs may continue to accumulate the effects of photochemical processes and re-develop cryoconite. The nutrient content of icebergs, nutrient distributions and their ratios might therefore not be static and in fact subject to semi-continuous changes. As ice moves downstream from ice sheets to the coastline, critical physical processes may exert a strong influence on the characteristics of the ice which ultimately calves into the ocean (Smith et al., 2019). At the base of floating ice tongues and ice shelves, the melt-rates of basal ice layers exposed to warm ocean waters can be rapid. Beneath the floating ice tongue of Nioghalvfjerdsbræ in northeast Greenland, for example, a melt rate of $8.6 \pm 1.4$ m year$^{-1}$ is likely sufficient to remove most sediment-rich basal ice prior to iceberg calving (Huhn et al., 2021). In other similar cases worldwide, calved ice may ultimately be deprived of basal layers which might otherwise have carried distinct labile sediment loadings reflecting subglacial processes (Smith et al., 2019). Nevertheless, post-calving the nutrient content of ice may still be strongly affected by 'new' ice-sediment interactions. Icebergs which become grounded, or scour shallow coastal sediments, may temporarily re-acquire a basal layer loaded with sediment (Gutt et al., 1996; Syvitski et al., 1987; Woodworth-Lynas et al., 1991). Scoured sediments may be physically and chemically distinct from those acquired from land-slides or basal glacial processes and thus also temporarily introduce different nutrient ratios and concentrations in ice and melt water (Forsch et al., 2021). Finally, whilst

many research questions concerning the effects of the cryosphere on the ocean relate to melting processes, marine ice formation is a mechanism via which ice growth can occur in the water column (Craven et al., 2009; Lewis and Perkin, 1986; Oerter et al., 1992). Marine ice is formed from supercooled seawater around Antarctica via the formation of platelet, or frazil, ice crystals. Whilst the chemical composition of this ice is poorly studied, measurements from the Amery Ice Shelf suggest marine ice has relatively high dissolved Fe (dFe) concentrations (e.g. 339-691 nM dFe, Herraiz-Borreguero et al., 2016). The origin of this dFe may be subglacial, potentially indicating a synergistic effect between subglacial and ice melt Fe sources. Similar synergistic effects have been suggested from model studies concerning sea ice and ice shelves, whereby sea ice may trap and later release Fe that originates from ice shelves (Person et al., 2021). A 'source-to-sink' narrative concerning iceberg-derived (micro)nutrient delivery from ice directly into the ocean may therefore be over-simplistic. It is important to recognise that the extent of spatial and temporal overlap between different (micro)nutrient sources may result in interactive effects in annual budgets. Such effects could arise due to the underlying physical processes and/or the seasonal timing of micro(nutrient) sources and sinks (Boyd et al., 2012; Person et al., 2021).

The (micro)nutrient content of icebergs and the associated fluxes of (micro)nutrients to the marine environment have been commented on around Greenland, Antarctica, and in smaller catchments around Svalbard (Cantoni et al., 2020; Nomura et al., 2023; Smith Jr. et al., 2007). Icebergs are widely thought to constitute a major source of Fe, particularly particulate Fe, to the ocean (Lin et al., 2011; Lin and Twining, 2012; Raiswell et al., 2016). We hypothesize that dMn, which shares similar sources with dFe, but is less susceptible to scavenging in the ocean, may also be delivered by icebergs with comparable annual fluxes to dFe. Several studies have also hinted at considerable dSi (up to 10 µM, Meire et al., 2016) or bioaccessible nitrogen concentrations (up to 5 µM) within ice (Parker et al., 1978; Vernet et al., 2011). Macronutrient concentrations in glacial ice are primarily hypothesized to reflect atmospheric deposition (Vernet et al., 2011), but it is unclear whether or not concentrations in calved ice largely reflect those originally deposited on ice sheet surfaces. The extent to which sediment incorporation into ice affects nutrient dynamics in ice melt also remains unclear. Are macronutrient and micronutrient concentrations in ice comparable at regional scales, or are there important regional differences due to

changes in basal ice layer thickness, sediment load, and sediment acquisition/loss processes in nearshore waters between regions? Calved ice from small marine-terminating glaciers in Svalbard, for example, can have extremely high sediment loads of up to 200 g $L^{-1}$ (Dowdeswell and Dowdeswell, 1989), compared to lower values of 0.6-1.2 g $L^{-1}$ in the Weddell Sea (Shaw et al., 2011). Are higher sediment loads also accompanied by increased concentrations of dissolved silica and trace metals in ice melt? Or, alternatively, is the loss of sediment from ice too fast, and any associated chemical weathering processes too slow, to significantly affect the composition of ice melt?

In order to evaluate whether or not there are regional differences in the (micro)nutrient content of icebergs and the associated fluxes into the ocean, here we assess the concentration of macronutrients ($NO_x^-$, dSi and $PO_4^{3-}$), micronutrients (dissolved Fe and Mn) and total dissolvable metals (Fe and Mn) from calved ice across multiple Arctic and Antarctic catchments. In order to investigate potential spatial and temporal biases associated with seasonal shifts and the general targeting of smaller ice fragments to collect samples, we include repeat samples from five campaigns in Nuup Kangerlua (a fjord hosting three marine-terminating glaciers in southwest Greenland) and a comparison of recently calved ice from inshore and offshore ice samples in Disko Bay (west Greenland). Throughout, we test the null hypothesis that icebergs from different regions have no differences in macronutrient or micronutrient (Fe and Mn) concentrations.

## 2 Methods

### 2.1 Sample collection

Iceberg samples were collected by hand or by using nylon nets to snag ice floating fragments. Sample collection was randomized at each field site location (Fig. 1 and Supp. Table 1) by collecting ice samples at regular intervals along pre-defined transects. 1–5 kg ice pieces were retained in low-density polyethylene (LDPE) bags and melted at room temperature. The first 3 aliquots of meltwater were discarded to rinse the LDPE bags. Meltwater was then syringe filtered (0.2 µm, polyvinyl difluoride, Millipore) into pre-cleaned 125 mL LDPE bottles for dissolved trace metal analysis and 20 mL polypropylene tubes for dissolved nutrient analysis. All plasticware for trace metal sample collection was

pre-cleaned using a three-stage protocol: detergent, 1 week soak in HCl (1 M reagent grade), and 1 week soak in HNO$_3$ (1 M reagent grade) with three deionized water rinses after each stage. Filters for trace metal analysis were pre-rinsed with HCl (1 M reagent grade) followed by deionized water. Some unfiltered samples were also retained for total dissolvable metal analysis.

In Disko Bay (west Greenland), a targeted exercise was conducted to test whether distinct regional patterns of ice nutrient concentrations could be associated with specific calving locations. During cruise GLICE (R/V Sanna, August 2022) ice collection was conducted as per other regions close to the outflow of Sermeq Kujalleq (also known as Jakobshavn Isbræ) and Eqip Sermia (Supp. Table 1). Additionally, ice fragments were collected from two large icebergs in Disko Bay, referred to herein as fragments from Iceberg "Beluga" and Iceberg "Narwhal". These icebergs were tracked using the ship's radar by logging the coordinates and relative bearing of the approximate centre of the iceberg at regular time intervals. In Nuup Kangerlua (southwest Greenland), samples were collected on 5 repeated campaigns spanning boreal spring and summer in different years (May 2014, July 2015, August 2018, May 2019 and September 2019) to assess the reproducibility of data from the same region by different teams deploying the same methods in different months and years.

## 2.2 Sediment load measurements

Wet sediment sub-samples were dried at 60°C to determine sediment load (dry weight of sediment per unit volume, mg L$^{-1}$). Sediment load was determined for a subset of randomly collected ice samples in parallel with (micro)nutrients in the Antarctic Peninsula. In Maxwell Bay (King George Island), a targeted exercise was conducted to collect ice with embedded sediment. Eight large ice fragments (10-45 kg) with sediment layers embedded within the ice were retained in sealed opaque plastic boxes. These fragments were specifically selected to avoid the possibility of including samples with surface sediment acquired by ice scouring the coastline or shallow sediments. Boxes were half-filled with seawater from the bay. Sediment-rich ice was left to melt in the dark with an air temperature of ~5-10°C. Periodically (after 2, 4, 8, 16, 24, and 48 hours) the water was weighed and settled sediment was removed by decanting and filtration before estimating its dry weight.

## 2.3 Chemical measurements

Dissolved trace metal samples were acidified after filtration to pH 1.9 by addition of 180 µL HCl (UPA, ROMIL) and allowed to stand upright for >6 months prior to analysis. Unfiltered trace metal samples were acidified similarly and trace metals in these samples are subsequently referred to as 'total dissolvable'; defined as dissolved metals plus any additional metals present which are soluble at pH 1.9 after 6 months of storage. Analysis via inductively-coupled, plasma mass spectrometry (ICP-MS, Element XR, ThermoFisher Scientific) was undertaken after dilution with indium-spiked 1 M $HNO_3$ (distilled in-house from SPA grade $HNO_3$, Roth). 4 mL aliquots of total dissolvable samples were filtered (0.2 µm, polyvinyl difluoride, Millipore) immediately prior to analysis.

Calibration for Fe and Mn was via standard addition with a linear peak response from 1–1000 nM ($R^2$ > 0.99). Analysis of the reference material CASS-6 yielded a Fe concentration of 26.6 ± 1.2 nM (certified 27.9 ± 2.1 nM) and a Mn concentration of 37.1 ± 0.83 nM (certified 40.4 ± 2.18 nM). Due to the very broad range of Fe concentrations in ice samples, samples were run using varying dilution factors. Precision is improved at low dilution factors so we report results from the lowest dilution factor that could be used to keep Fe and Mn concentrations within the calibrated range (in many cases dissolved samples could be run without dilution). Dissolved samples were initially run at a tenfold dilution, using 1 M $HNO_3$. A 1 M $HNO_3$ blank from the same acid batch was analysed every 10 samples and in triplicate at the start and end of each sample rack (90 × 4 mL sample vials). Total dissolvable samples (unfiltered, acidified samples) were initially run at a hundredfold dilution followed by a tenfold dilution for samples with nanomolar concentrations. Samples with measured concentrations of Fe or Mn <25 nM were then re-run without dilution. Detection limits, assessed as 3 standard deviations of blank (1 M $HNO_3$) measurements, varied between batches (and dilution factors) but were invariably <0.86 nM dFe and <0.83 nM dMn for the standard tenfold dilution analyses. The field blank (deionized water filtered and processed as a sample) was below the detection limit. As in a majority of cases samples were run by dilution, the 1 M $HNO_3$ acid used to both dilute samples and run as a reagent blank every 10 samples was therefore considered the

most useful blank measurement. Mean (±standard deviation) blank (1 M $HNO_3$) measurements varied by acid batch from $0.06 \pm 0.02$ nM dFe, $0.03 \pm 0.02$ nM dMn; to $0.38 \pm 0.08$ nM dFe, and $0.14 \pm 0.08$ nM dMn.

Where macronutrient samples were not collected in parallel with trace metals, samples preserved for trace metals were analysed for $PO_4^{3-}$ and dSi (this was not possible for $NO_x^-$ because of residual contamination from concentrated $HNO_3$ in LDPE bottles). Analysis of macronutrients was conducted for $NO_3^-$, $NO_2^-$, $PO_4^{3-}$ and dSi by segmented flow injection analysis using a QUAATRO (Seal Analytical) auto-analyzer (Hansen and Koroleff, 1999). Recoveries of a certified reference solution (KANSO, Japan) were $98 \pm 1\%$ $NO_x^-$, $99 \pm 1\%$ $PO_4^{3-}$ and $97 \pm 3\%$ dSi. Detection limits varied between sample batches and were $<0.10$ µM $NO_x^-$, $<0.02$ µM $NO_2^-$, $<0.10$ µM $PO_4^{3-}$, and $<0.25$ µM dSi.

**2.4 Data compilation**

In addition to new data from 367 new samples collected and analysed herein, existing comparable data was compiled from prior literature, most of which was processed in prior work by the same protocol in the same laboratories as herein (see Supp. Table 1). Inclusive of prior work, a total of 589 samples are available for interpretation (note that not all samples were analysed for all parameters so n varies between statistical analyses). Previously published data includes samples from Greenland, Svalbard, the Antarctic Peninsula, Patagonia and Iceland (De Baar et al., 1995; Campbell and Yeats, 1982; Forsch et al., 2021; Höfer et al., 2019; Hopwood et al., 2017, 2019; Lin et al., 2011; Loscher et al., 1997; Martin et al., 1990b). Altogether, 575 out of the 589 samples reported were collected and analysed as described herein at the same laboratories. Only 14 literature values were from other laboratories so there is a high degree of internal consistency in the methods used. Throughout concentrations are reported in units $L^{-1}$, referring to the concentration measured in meltwater.

**2.5 Statistical analysis**

To test if icebergs had statistically significant regional differences in (micro)nutrient concentrations depending on their origin at a hemisphere, regional or catchment scale, a multivariate PERMANOVA was realized (function adonis2 from vegan package, Oksanen et al., 2020) using the concentrations of trace metals (both dissolved and total dissolvable) and macronutrients ($NO_x^-$, $PO_4^{3-}$ and dSi). Along with this analysis a non-metric MultiDimensional Scaling (nMDS, function metaMDS from vegan package, Oksanen et al., 2020) was used to compute the ordination of the iceberg samples depending on their nutrient concentrations. An nMDS is an unconstrained ordination analysis that assess the similarities/dissimilarities among datapoints only using the set of variables informing the ordination (herein macro- and micronutrients concentrations). The variables considered for the analysis are summarized in orthogonal dimensions showing the more similar datapoints as closer (groupings of datapoints with similar characteristics) within the space created by the orthogonal dimensions. The same analyses were used to assess differences in Disko Bay samples collected in August 2022, in this case comparing iceberg samples collected in inshore and offshore zones. In both cases subsequent ANOVA (aov function package stats) and a Tuckey test (TukeyHSD function package stats) were undertaken to test for significant differences in specific (micro)nutrient concentrations.

The relationship between iceberg sediment load and the concentration of trace metals (both dissolved and total dissolvable) and macronutrients was determined by means of a linear regression (lm function package stats). For this analysis two outliers were removed from the dataset because their sediment load values were over an order of magnitude larger (50726 mg $L^{-1}$ and 6128 mg $L^{-1}$) than other values (total n=144); including these two data points would have disproportionately skewed the relationships. Finally, to analyse how melting and sediment release rates changed over time using the incubations in Maxwell Bay, we used the same procedure as Höfer et al., (2018). In short, we first tested if the relationship between melting and sediment release rates and time better fitted a linear or exponential relationship using a second-order logistic regression. Then, we tested the fit of the selected relationship (exponential in this case) to see if the relationship was significant and determined the percentage of variance explained (lm function package stats). Since the initial conditions of each incubation (i.e. iceberg size, shape and initial sediment load) varied, the rates for each individual experiment were normalized by dividing each rate by

the maximum rate registered in the same incubation. All statistical analyses and figures (package ggplot2)
were realized using R version 4.3.2 (R Core Team, 2023).
**3 Results**
**3.1 Nutrient distributions in the global iceberg dataset**
A total of 589 ice fragments have been analysed to date. The combined data is more balanced compared
to prior work in terms of coverage of Antarctica (45% of samples), Greenland (42% of samples), Svalbard
(8.1% of samples), and smaller sub-polar catchments in Patagonia, Canada, and Iceland (4.2% of
samples). There are however still some spatial biases in the data. Notably samples from Greenland are
largely from the west (Fig. 1), and samples from Antarctica are all from the Antarctic Peninsula or
downstream waters along the "Iceberg Alley" in the Weddell Sea and the South Atlantic sector of the
Southern Ocean (Tournadre et al., 2016). Almost all samples were collected in summer, with only a subset
of samples (from Nuup Kangerlua, Supp. Table 4) collected in spring and autumn to investigate potential
seasonal changes. At the catchment scale, Nuup Kangerlua (southwest Greenland, also known as
Godthåbsfjord, 15% of the dataset), Eqip Sermia (west Greenland, 11% of the dataset), Thunder Bay
(Western Antarctic Peninsula, 10% of the dataset), Kongsfjorden (Svalbard, 8.2% of the dataset), Disko
Bay (west Greenland, 5.1% of the dataset), and Nelson Island (Northern Antarctic Peninsula, 5.1% of the
dataset) are particularly well represented. The other 23 catchments each account for <5% of the samples.




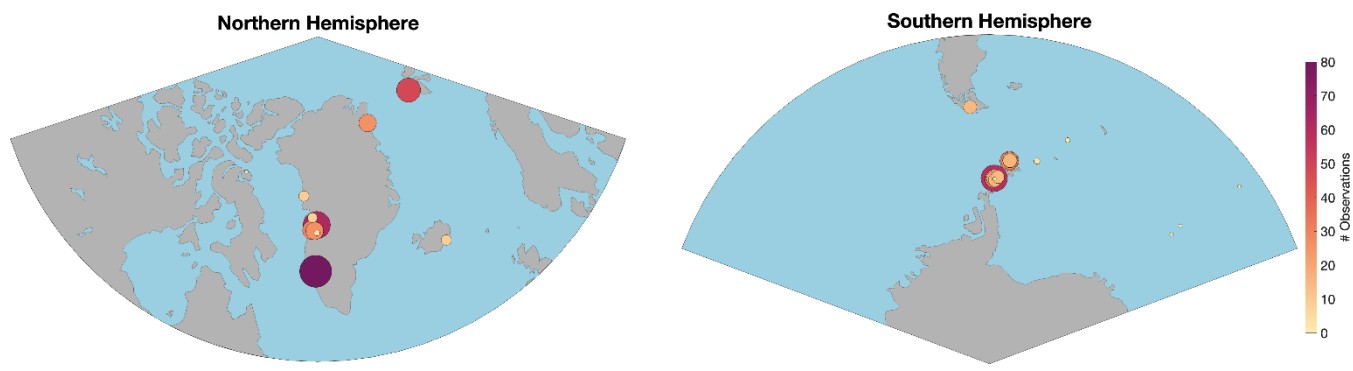

Figure 1. Sample distributions in the Northern and Southern Hemispheres. Literature values from prior work are included (see Supp. Table 1 for a full list of details).

Average macronutrient concentrations in ice samples were low with median concentrations of 0.04 µM $PO_4^{3-}$, 0.54 µM $NO_3^-$ and 0.02 µM dSi. Throughout the dataset $NO_2^-$ was close to, or below, detection thus $NO_3^-$ and $NO_x^-$ concentrations were practically identical with $NO_2^-$ almost invariably constituting <10% of $NO_x^-$ (mean 1.8%). Mean nutrient concentrations in all cases were higher than median concentrations and the large relative standard deviations indicated that variability between samples might mask any regional differences. Preliminary analysis revealed a large fraction of data below detection (i.e. concentrations <LOD) for several components particularly $PO_4^{3-}$ (24% of all measurements <LOD) and dSi (48% if all measurements <LOD). Other (micro)nutrients were less affected by detection limits, e.g. only 8% of $NO_x$ concentrations were <LOD. In any dataset with a large fraction of data <LOD, how these values are treated makes some difference to calculated statistics so reported averages vary for $PO_4^{3-}$ and dSi depending on how LOD values are treated. Removing values <LOD entirely would skew the statistical analyses. For example, the median values reported above increase from 0.04 to 0.05 µM $PO_4^{3-}$, and 0.02 to 0.19 µM dSi if values <LOD are excluded. For consistency throughout all statistical analyses, a value of '0' was therefore used to represent LOD data.

It has been previously reported that both TdFe and dFe concentrations are extremely variable within ice samples collected at the same location (Hopwood et al., 2017; Lin et al., 2011). This remained the case

with the expanded dataset herein with notable differences between the mean (82 nM dFe, 13 µM TdFe) and median concentrations (12 nM dFe, 220 nM TdFe) on a global scale. An extremely broad range of concentrations was also observed for both dissolved Mn (mean 26 nM, median 2.6 nM) and total dissolvable Mn (TdMn; mean 150 nM, median 10 nM). As per Fe, this reflected the skewed distribution of the dataset towards a low number of samples with extremely high concentrations. The highest 2% of TdMn samples accounted for 79% of the cumulative TdMn measured. Similarly, the highest 2% of TdFe samples accounted for 77% of the cumulative TdFe measured. Accordingly, there were very high relative standard deviations for both mean dMn ($26 \pm 160$ nM) and TdMn ($150 \pm 1500$ nM) which, as per Fe, remained high when data was grouped by region or catchment. Considering all (micro)nutrients measured, there were no significant differences in the iceberg chemical composition at a hemispheric (p value = 0.16) or regional (p value = 0.16) level. However, a PERMANOVA analysis showed significant differences ($R^2 = 0.24$, p value <0.001) at a catchment level. Similarly, an nMDS analysis (stress = 0.07) showed that samples from the same catchment tended to be grouped closer together (Fig. 2) and in general Antarctic samples were distributed on the left side, whereas Arctic samples were more abundant on the right side of the ordination analysis (Fig. 2).

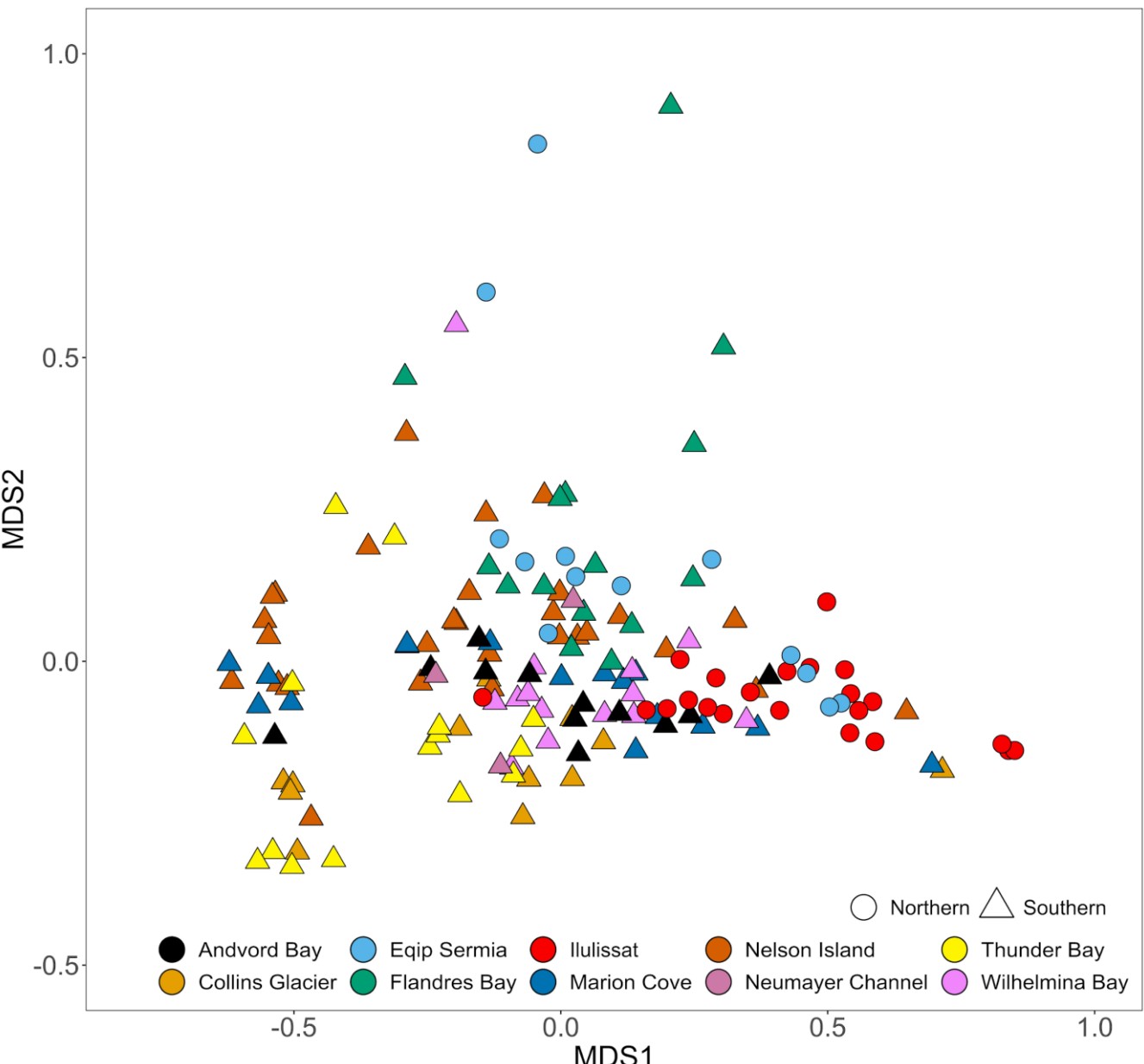

335

Figure 2. A scatter plot showing the results of an nMDS ordination analysis using macro- and micronutrient concentrations. Only samples with complete data for the following parameters are shown: $NO_x^-$, $PO_4^{3-}$, dSi, dFe, TdFe, dMn and TdMn. A non-metric MultiDimensional Scaling (nMDS) ordination is used to represent multi-dimensional data in a reduced number of dimensions. MDS1 and MDS2 are multidimensional scaling factors which represent the dissimilarities between the data sorted to catchment

level. Datapoints represent individual samples. Datapoints which appear further apart are more different, whereas those that cluster together are more similar. A PERMANOVA analysis of iceberg nutrient concentrations showed significant differences at a catchment level ($R^2 = 0.24$, p value <0.001). Shapes denote hemispheres, while colours denote specific sampling locations.

The ratio of TdFe:TdMn was linear ($R^2 = 0.95$, calculated excluding the highest 2% of Mn and Fe concentrations to avoid skewing the gradient, Fig. 3). Furthermore, the total dissolvable Mn:Fe ratio of 0.0225 (linear regression TdMn = $0.0225 \times$ [TdFe]) was close to mean continental crust composition which is approximately 0.1% MnO and 5.04% FeO by weight (producing a ratio of 0.020) (Rudnick and Gao, 2004). In contrast, no clear relationship was observed between dFe and dMn. For all data, all Antarctic data and all Greenlandic data, respectively, the mean dMn:dFe (0.47, 0.50 and 0.28) and median dMn:dFe (0.17, 0.19 and 0.11) ratios were however consistently higher than the TdMn:TdFe ratio. This indicates an excess of dMn compared to the lithogenic ratio observed in the total dissolvable fraction.

Neither dMn or dFe correlated well with dSi. Throughout the whole dataset, dSi concentrations were low. Only 7 of 478 samples had dSi concentrations >10 µM, only 9.4% of samples had concentrations >1.0 µM, and 48% of all samples were below detection. Dissolved Si therefore had concentrations and a distribution much more like $NO_x^-$ and $PO_4^{3-}$ than Mn or Fe. This was not typically the case in glacier runoff close to the sites where ice was collected (Supp. Table 2). With the exception of subglacial runoff collected on Doumer Island (South Bay, Western Antarctic Peninsula), dSi concentrations in runoff were always high relative to both nitrate in runoff (typically $\sim 12 \times$ [$NO_x^-$]) and to the mean dSi concentration in icebergs. Doumer Island consists of a small ice cap which is likely cold-based with steep topography, such that runoff-sediment interaction is likely limited.

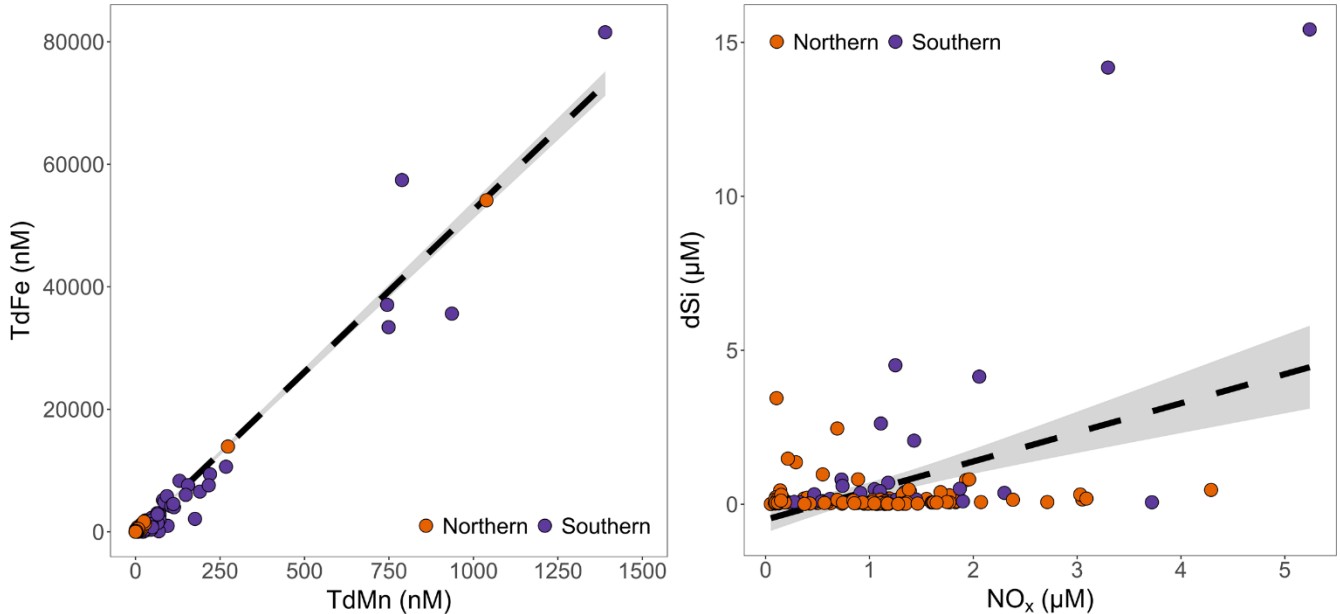

Figure 3. A comparison of (micro)nutrient concentrations in all ice fragments where concentrations were above the detection limits. *Left* Total dissolvable Fe and total dissolvable Mn were strongly correlated (p value <0.001, $R^2 = 0.95$), note the highest 2% of measured concentrations were excluded to avoid skewing the gradient. *Right* dSi and $NO_x^-$ had a weak correlation (p value <0.001, $R^2 = 0.19$). The 95% confidence interval is shaded in grey.

No significant relationship was evident between $PO_4^{3-}$ and $NO_x^-$ concentrations, whereas a weak, but significant, relationship was evident between dSi and $NO_x^-$ concentrations (Fig. 3). A subset of samples appeared to show a close to 1:1 relationship between dSi and $NO_x^-$, which resembles the Redfield Ratio (Redfield, 1934). A closer inspection of these points shows they accounted for about 14% of the sub-dataset where all macronutrient concentrations were detectable (n=22 for those with $[NO_x^-]$ and $[dSi] >0.4$ µM, for lower concentrations it is largely arbitrary determining whether or not samples can be assigned to the group). Samples in this group include multiple catchments but with a large component from Ilulissat (32% of datapoints) and Nuup Kangerlua (55% of datapoints), both of which were over-represented compared to their proportional importance in the sub-dataset where they each constituted 18% of datapoints. Antarctic samples and samples from Eqip Sermia were under-represented in this ~1:1 group,

accounting for 0 and 2 (9%) samples, respectively, despite contributing 26% and 20% of the samples with all macronutrients detectable. The ~1:1 datapoints all refer to summertime so cannot easily be explained as mistaken sea ice samples. Furthermore, observed nutrient concentrations were often too high to be explained by carry-over from seawater contamination (see Section 3.2). The ratios of dSi: $NO_3^-$ also did not consistently match the ratio in near-surface fjord water samples where this was collected in parallel with icebergs. Whilst the dSi: $NO_3^-$ ratio in most near-surface samples from the Ilulissat Icefjord in August 2022 was ~1 ($1.39 \pm 0.61$, n=25 in August 2022), for Nuup Kangerlua in August and September 2019 the ratio of dSi: $NO_3^-$ was always >18 (Krause et al., 2021). A ~1:1 $NO_x$:dSi ratio in ice nevertheless resembles a marine origin.

## 3.2 Evaluating reproducibility and potential sampling biases

Glacial ice can usually be visually distinguished from sea ice due to its distinct texture, colour and morphology. For meltwater samples that were tested for salinity, values were always <0.3 psu. However, even minor traces of seawater in samples would be sufficient to impart a measurable macronutrient concentration change because ice macronutrient concentrations were generally very low compared to pelagic macronutrient concentrations in the corresponding sampling regions. This is particularly the case at the Antarctic sample sites where high macronutrient concentrations of 20-80 µM dSi, 1-2 µM $PO_4^{3-}$ and 10-30 µM $NO_3^-$ are relatively typical of marine waters (e.g. Höfer et al., 2019; Forsch et al., 2021; Trefault et al., 2021). Close to marine-terminating glaciers in the Arctic, macronutrient concentrations in near-surface waters can still be elevated relative to the low concentrations reported for ice, e.g. 1-30 µM dSi, 0.2-0.7 µM $PO_4^{3-}$ and 0-10 µM $NO_3^-$ for the inner part of Nuup Kangerlua (Krause et al., 2021; Meire et al., 2017). Thus, seawater macronutrient concentrations were generally equal to, or greater than ice concentrations at the locations where ice calves.

Using the maximum observed marine macronutrient concentrations for our Antarctic sampling locations, assuming no detectable macronutrients in ice and that salinity of 0.3 exclusively reflected the carry-over of seawater from sampling, nutrient concentrations of up to 0.26 µM $NO_3^-$, 0.02 $PO_4^{3-}$ µM and 0.069 µM dSi could be observed as a seawater contamination signal. The rinsing procedure used to collect samples

herein whereby ice was sequentially melted, with the meltwater then used to swill and rinse the sample bag, was designed precisely to minimize trace metal contamination and three such rinses undertaken correctly would theoretically remove ~99.99% of any saline water collected with an ice sample in addition to any contamination from ice handling. This would also not leave a detectable (>0.01) salinity increase in the collected sample such that any detected salinity would have to come from ice melt. Sea ice samples were not targeted for sampling herein, but two samples were collected during the 2017 Pia fjord campaign (Patagonia) alongside calved ice samples and measured macronutrient concentrations were: 2.00 and 5.97 $\mu M$ $NO_x^-$, 0.08 and 0.13 $\mu M$ $PO_4^{3-}$, 0.28 and 0.63 $\mu M$ dSi. These sea ice $NO_x^-$ and dSi concentrations were above average compared to freshwater ice samples collected in the same location (Supp. Table 2). Similarly, samples of land fast sea ice from Antarctica generally have high concentrations of all macronutrients compared to iceberg samples reported herein (Grotti et al., 2005; Günther and Dieckmann, 1999; Nomura et al., 2023). It is apparent from the ratio of $NO_x^-$: $PO_4^{3-}$:dSi in sea ice that the high nutrient concentration in sea ice have a saline origin (Henley et al., 2023). Sampling protocols for sea ice are however different in several aspects particularly the application of a sequential rinsing (for glacial ice, but not for sea ice) and ambient temperatures during sample collection. A sequential rinsing with sea ice, as applied herein, might lead to an uneven distribution of nutrients in meltwater samples due to the layered structure of sea ice and the effects of brine channels (Ackley and Sullivan, 1994; Gleitz et al., 1995; Vancoppenolle et al., 2010). With the possible exceptions of regions that experience ice mélange (a mixture of sea ice and icebergs) and/or marine ice, glacial ice is expected to be more homogenous with respect to salinity. In prior work we also demonstrated no sustained trend in Fe concentrations when aliquots of meltwater were collected from ice fragments in series (Hopwood et al., 2016). A further critical difference with sea ice samples concerns ambient conditions as all ice samples collected herein were obtained from seawater with temperatures >0°C i.e. under conditions where ice was melting when it was collected. Conversely, a large fraction of sea ice cores studied to date refer to conditions without *in situ* melt occurring (Henley et al., 2023).

During the dedicated iceberg cruise campaign GLICE in Disko Bay (August 2022), ice collection was confined to 4 subregions of interest (Fig. 4, Supp. Table 3). There was partial ice cover in Disko Bay

during boreal summer, which was mainly limited to a patch of high iceberg density close to the outflow of Ilulissat Icefjord. Combined with the confined nature of the coastal fjords sampled and the relatively fast disintegration of smaller ice fragments, it was possible to identify with a high degree of certainty the origin of ice within each subregion (Fig. 4). Within the fjord system hosting the marine-terminating glacier Eqip Sermia, ice fragments were highly likely to have originated from either Eqip Sermia itself or, if not, from adjacent calving fronts in the same fjord. Similarly, close to the outflow of Ilulissat Icefjord, ice fragments were highly likely to have originated from Sermeq Kujalleq. Ice slicks which were visibly observed to calve from two offshore icebergs within an hour prior to sample collection each constituted an additional subregion of interest. The two icebergs, referred to herein as 'Narwhal' and 'Beluga' were both isolated from other floating ice features with maximum dimensions above the waterline of >100 m width and >20 m height (Fig. 4). Radar measurements determined that 'Narwhal' was approximately stationary throughout the observation period (~12 hours) likely pirouetting on an area of shallow bathymetry. Iceberg 'Beluga' was free-floating and proceeding northwards along a trajectory through the area which hosted the highest observed iceberg densities in Disko Bay over the cruise duration (mid-August 2022).

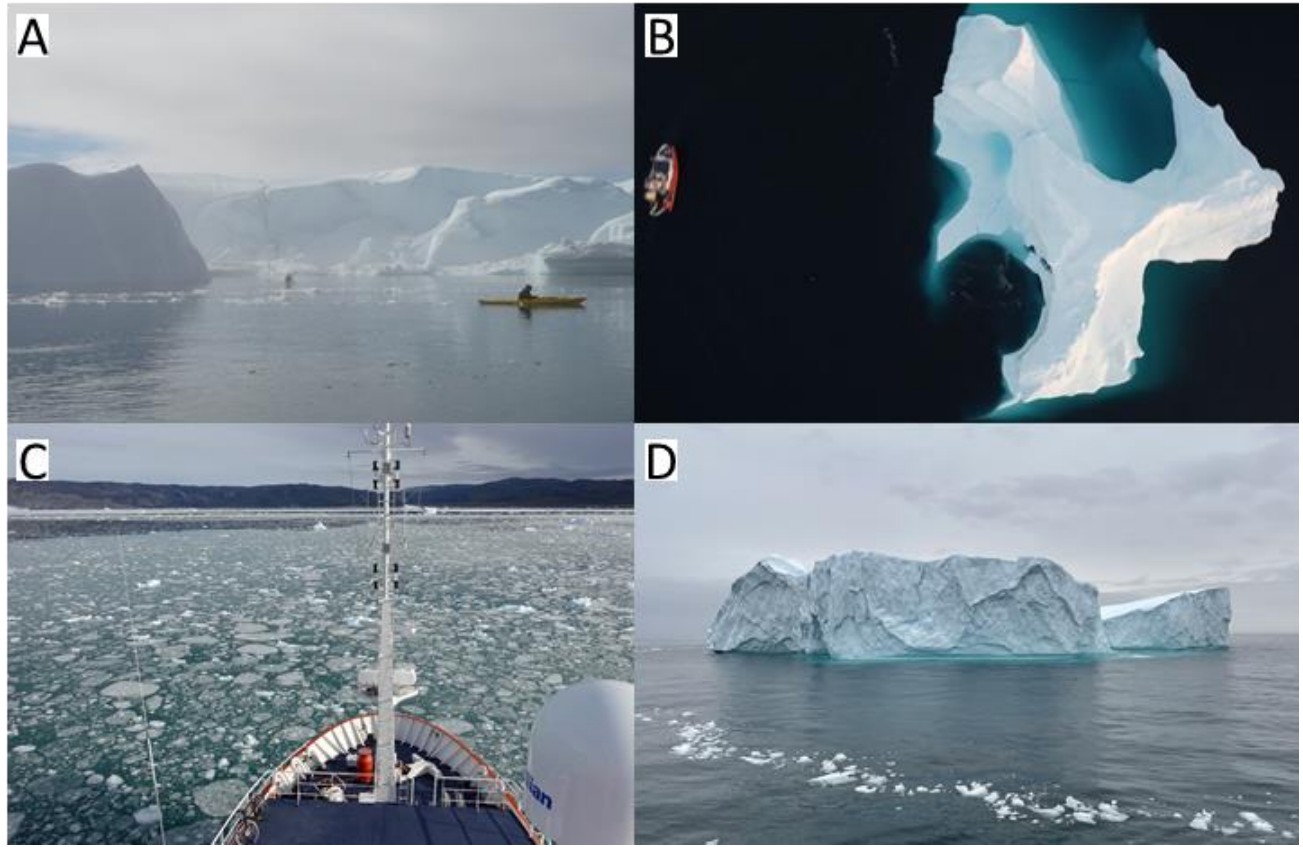


Figure 4. Ice sample collection areas in four distinct regions of Disko Bay. A Icebergs grounded on the
sill at the entrance to the Ilulissat Icefjord. B An offshore iceberg which was grounded during the sampling
period referred to herein as iceberg 'Narwhal'. C Ice fragments in front of the marine-terminating glacier
Eqip Sermia. D An offshore iceberg which was free-floating during the sampling period referred to herein
as iceberg 'Beluga'.

Ice from the 4 sampled subregions in Disko Bay was similar in all cases with overlapping ranges for the
$NO_x^-$, $PO_4^{3-}$ and dSi concentrations of ice at different locations (Fig. 5). A PERMANOVA analysis
showed small, but significant, differences ($R^2 = 0.15$, p value = 0.002) in the chemical composition of
iceberg samples collected inshore (Groups A and C, Fig. 4) or offshore (Groups B and D, Fig. 4) in Disko
Bay when combining groups. An ordination analysis (nMDS stress = 0.04) showed that offshore icebergs
were grouped together on the left side of the ordination, whereas inshore icebergs were more common on

the right side of the ordination (Fig. 5). In general, offshore and inshore icebergs presented similar concentrations of all nutrients in most of the samples, except for a few inshore samples that had higher concentrations of all nutrients (Fig. 5). When testing these differences for each individual nutrient, only $PO_4^{3-}$ showed significant differences between the two categories (p value = 0.035), with offshore icebergs showing lower concentrations (Fig. 5). The difference between inshore and offshore ice, whilst present, was therefore relatively modest.

Further insight can be gained from a comparison of all data available from Nuup Kangerlua, a relatively well-studied glacier fjord in southwest Greenland. The fjord hosts three marine-terminating glaciers with heavy ice mélange cover observed in the inner fjord year-round and some sea ice in the inner fjord during winter. Samples were collected from the fjord during five independent field campaigns from 2014 to 2019 in different seasons from May in boreal spring to September in boreal autumn. Considering the number of parameters sampled and the relatively high standard deviation of almost all parameters relative to the mean or median measured concentrations, there was limited evidence for any seasonal or inter-campaign differences (Supp. Table 4). No significant differences (p>0.05) were found between groups of samples obtained at the same field site when organizing the complete dataset by field site and defining each separate field campaign as a group.

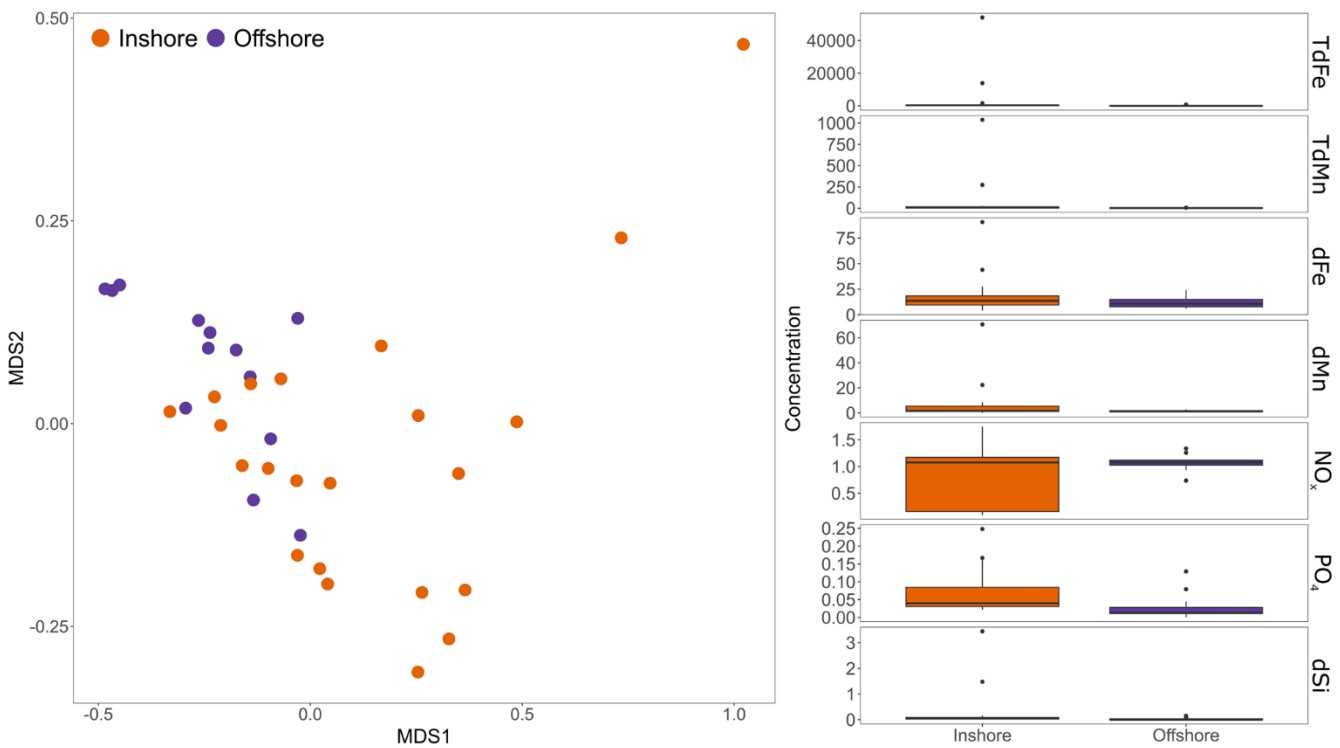

481

Figure 5. Comparison of nutrient concentrations from inshore and offshore ice samples collected in Disko Bay (August 2022, see Fig. 4). *Left* An ordination analysis (nMDS) comparing concentrations of all nutrients measured in ice contrasting inshore and offshore areas of Disko Bay. A PERMANOVA analysis of iceberg nutrient concentrations showed weak but significant differences between both areas ($R^2 = 0.15$, p value = 0.002). *Right* A direct comparison of all nutrient concentrations for the same dataset. Units: $\mu M$ for dSi, $NO_x^-$ and $PO_4^{3-}$; nM for all trace metals. Only $PO_4^{3-}$ showed a significant difference between the two categories (p value = 0.035).

### 3.3 Sediment load within icebergs and its relationship with nutrient concentration

The sediment load within icebergs collected around the Antarctic Peninsula was highly variable with a maximum of 5072 mg $L^{-1}$ and a minimum of 0.69 mg $L^{-1}$ (median 8.5 mg $L^{-1}$ and mean 430.5 mg $L^{-1}$). Particle loads were assessed in three Antarctic locations. The median dry mass was similar across three areas, but the mean (± standard deviation) dry mass was more variable due to the occasional sample with a high sediment load. Mean dry masses across three areas were: Maxwell Bay, King George Island, (n=65)

910 ± 6300 mg L$^{-1}$; Thunder Bay and Neumayer Channel, Wiencke Island, (n=19) 35 ± 110 mg L$^{-1}$; and South Bay, Doumer Island, (n=60) 39 ± 98 mg L$^{-1}$. Median sediment loads in the three regions were 12, 2.5 and 7.7 mg L$^{-1}$, respectively. The heterogeneous distribution of sediments was reflected in the fact that ~2% of samples collected contributed ~90% of the total sediment retrieved from the iceberg samples collectively (Fig. 6). This distribution is similar to previous analysis regarding TdFe (Hopwood et al., 2019), and sediment load in icebergs from Svalbard (Hopwood et al., 2017). It also qualitatively matches the distribution of TdMn and TdFe observed herein (see Section 3.1).

As Fe, Mn and dSi might have sedimentary origins, we tested if there were any significant relationships between the sediment load of an iceberg and the concentration of each macronutrient and both total dissolvable and dissolved trace metals (Fig. 6). For $NO_x^-$ and $PO_4^{3-}$ there was no significant relationship between sediment load and concentration (p values of 0.18 and 0.26 respectively). This is consistent with the hypothesis that these nutrients primarily have an atmospheric deposition origin which contributes only a minor fraction of the sediment load to bulk ice. Conversely, TdFe, TdMn, dFe, dMn and dSi all had significant relationships with sediment load. The concentrations of the total dissolvable fraction of trace metals showed better fits (TdFe $R^2$ = 0.43, p value <0.001; TdMn $R^2$ = 0.43, p value <0.001), than the dissolved phases of metals (dFe $R^2$ = 0.30, p value <0.001; dMn $R^2$ = 0.20, p value <0.001) and dSi ($R^2$ = 0.28, p value <0.001). This is consistent with the expectation that englacial sediment drives a direct enrichment in TdFe and TdMn, which increase proportionately with sediment load. The enrichment of dFe, dMn and dSi is more variable and may depend on the specific conditions that sediment and ice experience between englacial sediment incorporation and sample collection.

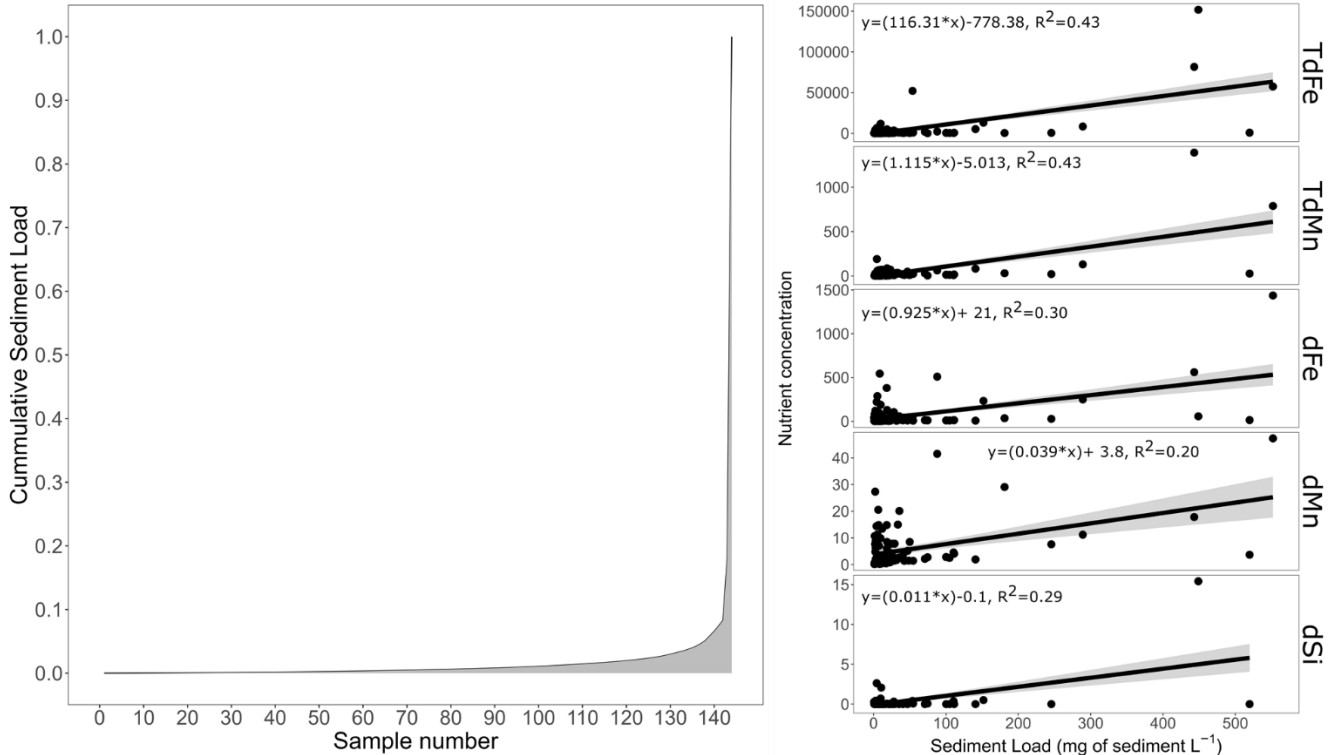


Figure 6. Iceberg sediment load and its relationship with nutrient concentrations. *Left* The uneven release
of sediment in randomly collected ice samples from the Antarctic Peninsula. *Right* The relationship
between nutrient concentrations and sediment load for ice samples from the Antarctic Peninsula (no
samples from elsewhere determined sediment load on the same ice fragments as nutrient concentrations).
Only significant (p value <0.001) relationships are shown. No significant relationship was evident for
sediment load with nitrate or phosphate. Units: μM for dSi, nM for all trace metals.

On several occasions in Nuup Kangerlua and Maxwell Bay we observed structures up to several
centimetres wide/deep on iceberg surfaces akin to cryoconite holes both above and below the waterline.
The sediment within such holes was easily disturbed when approaching ice fragments. The regular
agitation and movement of floating ice fragments and the chaotic nature of calving events suggests that
cryoconite holes on icebergs formed *in situ* rather than being relics of a glacier surface prior to calving.
This raises an interesting question about whether sediment-rich layers and any associated nutrients could
be subject to disintegration mechanisms distinct from bulk ice. When large ice samples weighing 10-45
kg were stored in the dark at 5-10°C, higher loads of sediment were released in the initial melt fractions
(Supp. Fig. 1). This trend was highly reproducible occurring in all observed experiments (n=8) when large
ice samples specifically targeted for their high englacial sediment loads were retained. The sediment
release rate declined with an exponential logarithmic function over the first 48 hours (Supp. Figure 1). It
should be noted that randomly collected samples had much lower sediment loads (Fig. 6).
**4 Discussion**
**4.1 Insights into nutrient origins from ratios**
There are several distinct mechanisms via which ice could accumulate different nutrient ratios.
Precipitation and aerosol on ice surfaces would be expected to deposit $NO_x^-$ and $PO_4^{3-}$ (Fischer et al.,
1998; Kjær et al., 2015), assuming a limited biogeochemical imprint from surface biological (or
photochemical) processes. Atmospheric deposition of $NO_x^-$ and $PO_4^{3-}$ varies regionally. Snow $NO_3^-$
deposition over central Greenland is reported as $1.21 \pm 0.19$ µmol kg$^{-1}$ for recent and $0.56 \pm 0.19$ µmol
kg$^{-1}$ for pre-industrial values (Fischer et al., 1998). Reported concentrations of $PO_4^{3-}$ are more sensitive
to the method used due to universally low concentrations. Phosphate concentrations in ice from the last
glacial period in Greenland are reported to range from 3 to 62 nM (Kjær et al., 2015). These ranges are
similar to the $NO_3^-$ and $PO_4^{3-}$ values we report for Greenlandic calved ice herein: mean ($\pm$ standard
deviation) $0.78 \pm 0.69$ $NO_3^-$, median $0.74$ $NO_3^-$, mean $36 \pm 50$ nM $PO_4^{3-}$, and median 28 nM $PO_4^{3-}$. Modern
atmospheric deposition is expected to impact the N:P ratio as atmospheric pollution is generally
associated with higher N:P ratios (e.g. Peñuelas et al., 2012) and could explain the increase in N:P ratio
at higher $NO_3^-$ concentrations. Atmospheric deposition of $NO_3^-$ is Antarctica is less directly affected by
anthropogenic emissions, but the ranges of $NO_3^-$ reported for snow and ice samples overlap with the
corresponding values for Greenland e.g. ranges of 0.08-2.12 µM (Akers et al., 2022) and 0.29-2.58 µM
(Neubauer & Heumann., 1988).

In addition to an atmospheric deposition signal in ice macronutrient concentrations for $NO_x^-$ and $PO_4^{3-}$,
some degree of sedimentary signal might also affect dSi concentrations due to release of dSi from glacier-

associated weathering processes (Halbach et al., 2019; Wadham et al., 2010). In contrast no, or very limited, release of $NO_3^-$ or $PO_4^{3-}$ is expected from weathering which is supported by the correlations herein (Fig. 6). Sediment associated with an iceberg could be from basal layers, other englacial sediment entrained prior to calving, or acquired from scouring events subsequent to calving. Shallow areas of all field sites herein had grounded icebergs. In Disko Bay during 2 weeks of cruise observations in August 2022 for example, the majority of large (>100 m width above water line) icebergs were observed to be grounded. In terms of TdFe, TdMn, dFe, dMn and dSi we hypothesize that two categories of sediment may be distinguishable. Englacial sediment with little biogeochemical processing should retain a TdFe:TdMn ratio which is close to the crustal abundance ratio of Fe:Mn, with low dFe, dMn and dSi concentrations. Basal sediment layers, particularly from catchments with warm-based glaciers, may have a similar TdFe:TdMn ratio but higher concentrations of dFe, dMn and dSi due to more active biogeochemical processing in subglacial environments (e.g. Wadham et al., 2010; Tranter et al., 2005). Finally, scoured sediments acquired after calving could constitute a broad range of compositions considering the gradient in benthic conditions along glacier fjords (Laufer-Meiser et al., 2021; Wehrmann et al., 2013) and may accordingly contain more biogenic and/or authigenic phases than englacial sediment. These sediments may be highly variable in composition but should impart high TdFe and TdMn concentrations, with varying Fe:Mn ratios, and high dFe, dMn and dSi concentrations. Basal sediments and scoured sediments from fjord environments therefore probably cannot be distinguished unambiguously from concentrations measured herein alone. Yet we can likely distinguish englacial sediment from basal or scoured sediment. Dissolved Si concentrations were low across the whole dataset, suggesting basal ice was a very small component of sampled ice. The linear relationship between TdFe and TdMn across a wide range of observed concentrations also suggests minimal incorporation of authigenic mineral phases and, in combination with low dSi, hints that basal ice from warm-based glaciers is largely absent from this dataset. This is consistent with the expectation that basal layers are lost prior to, or rapidly following, iceberg calving (Smith et al., 2019). In contrast, in runoff sampled close to iceberg sampling regions, dSi concentrations were elevated (range 1.2-44 µM) and often considerably higher than concentrations measured in ice melt (Supp. Table 2).

The weak, but significant, relationships with dSi, dFe, dMn and sediment load; and the stronger relationships between TdFe and TdMn and sediment load are consistent both with a sedimentary origin of these components and the caveats that further physical and/or biogeochemical processing mechanisms have to be considered to fully explain the distributions of dSi, dFe and dMn (Fig. 6). As the concentrations of $NO_x^-$ and $PO_4^{3-}$ were consistent with an atmospheric origin, a varying concentration of dSi from sedimentary sources could also easily explain the observed trend in the $NO_x^-$:dSi and $PO_4^{3-}$:dSi ratios. Whilst elevated dFe and dMn concentrations in runoff reflect release of these phases from glacier-derived sediments (Hawkings et al., 2020; Raiswell, 2011), the concentrations herein for ice melt were not strongly correlated with each other or sediment load (Fig. 6). This could reflect the origin of dissolved Fe and Mn from distinct, different mineral phases, yet dFe concentrations generally correlate poorly with other trace elements in aquatic environments due to rapid scavenging onto particle surfaces and rapid aggregation of colloids (which are included within the '<0.2 µm' definition of dissolved herein) (Zhang et al., 2015). A poor correlation could also therefore reflect the tendency for inorganic dFe species to become rapidly scavenged close to source (Lippiatt et al., 2010). Measured concentrations herein refer to freshly collected meltwater so it is difficult to establish how dFe concentrations may have changed during the ice melting process. Conversely, dMn species are more stable in solution, especially in the photic zone (Sunda et al., 1983; Sunda and Huntsman, 1988), and this is often reflected in much higher dMn:dFe ratios in proglacial aquatic environments than would be expected based on crustal abundances (e.g. van Genuchten et al., 2022; Hawkings et al., 2020; Yang et al., 2022). Curiously, dSi also correlated poorly with all metal phases. This again could simply reflect different mineral phases driving elevated dSi, dFe and dMn concentrations (van Genuchten et al., 2022). Yet considering all of these elements are expected to be released from labile phases present in glacier-derived sediments, at least within specific regions some degree of correlation might be expected. Further work to quantify the rates of gross and net dFe, dMn and dSi release under *in situ* conditions within ice and frozen sediment layers, could perhaps elucidate processes via which net release of these components may be uncoupled. Photochemical processes are particularly likely to affect Fe and Mn release (Kim et al., 2010; Kim et al., 2024), and the scavenging potential of Mn and Fe species (van Genuchten et al., 2022) may also be important in terms

of how they interact with other dissolved and particulate components of the ice-sediment-meltwater matrix.

**4.2 Key role of sediment-rich layers, and their disintegration, for nutrient release**

Several works have speculated that Arctic and Antarctic icebergs may have distinct differences in sediment load, with the former generally having higher sediment loads than the later (Anderson et al., 1980). However, there are several observer biases in making such comparisons. Arctic icebergs are generally smaller due to the prevalence of tidewater glacier-derived ice rather than large ice shelves. Furthermore, due to the much easier logistical situation for observers in the Arctic, Arctic icebergs are more easily observed in coastal environments than Antarctic icebergs. Ice observed at a distance often appears cleaner than is the case upon closer inspection where sediment layers can be better identified. Nevertheless, a comparison of smaller ice fragments from Kongsfjorden in Svalbard and three localities in the Antarctic Peninsula showed that the former had higher sediment loads. Mean sediment loads of 21 g $L^{-1}$ (median 0.58 g $L^{-1}$) were previously reported for Kongsfjorden (Hopwood et al., 2019). Average sediment load values for ice fragments handled similarly from the Antarctic Peninsula were 8.5 mg $L^{-1}$ (median) and mean 430.5 mg $L^{-1}$ (mean), respectively, which are considerably lower. Contrasting warm/cold-based glaciers and the higher exposed land/ice cover ratio of the coastal glaciated Arctic may explain much of this difference.

Sediment-rich layers within icebergs have long been hypothesized to be particularly important for the delivery of the micronutrient Fe into the ocean (Hart, 1934) and this has been explicitly confirmed with measurements of dFe and particulate Fe (Lin et al., 2011; Raiswell, 2011). We verify herein, that sediment distribution is a major factor explaining TdFe and TdMn distribution, yet suggest this is a less important factor in explaining dFe, dMn and dSi distribution in icebergs (Fig. 5). The dynamics of sediment-rich layers and their fate in the marine environment is of special interest for trace metal biogeochemistry given the (co)-limiting role these micronutrients have for phytoplankton growth in the Southern Ocean (Hawco et al., 2022; Martin et al., 1990b). Yet multiple factors are likely important for determining the delivery of dFe and dMn to the marine environment because these fluxes do not simply scale with sediment input

as per TdFe and TdMn. A close association of TdFe and TdMn is perhaps unsurprising and corroborates a lithogenic origin for the vast majority of Fe present in icebergs. It also suggests limited biogeochemical processing of englacial material and/or rapid loss of basal ice layers preventing the modification of a lithogenic ratio in-between sediment acquisition by icebergs and sediment release in the ocean (Forsch et al., 2021).

A curious observation herein was that cryoconite formation was observed on ice fragments suggesting that, as is the case on glacier surfaces (Cook et al., 2015; Rozwalak et al., 2022), this can be an important process affecting sediment dynamics on icebergs. The unstable nature of icebergs, especially smaller icebergs, means that cryoconite holes are likely to be shorter lived than their glacier counterparts, but they still may constitute an important feature via which iceberg embedded sediment is processed. The accumulation of sediment as cryoconite could for example impede photochemical processing of particles, but also potentially create micro-gradients in $O_2$, pH and temperature that result in different chemical conditions than if particles were homogenously distributed (Rozwalak et al., 2022). On larger, more stable tabular icebergs, cryoconite may facilitate the growth of attached diatoms (Ferrario et al., 2012; Robison et al., 2011). These processes are well described on glacier surfaces but a critical difference in interpreting their significance in iceberg environments is that iceberg movement and rolling is likely to prevent the long-term development of cryoconite on anything other than large tabular icebergs. Nevertheless, the observation of such holes at centimetre size in environments where icebergs are free floating and rapidly disintegrating suggests that they might constitute an underappreciated mechanism of iceberg melt and sediment processing.

A further, to our knowledge, novel observation was the tendency of embedded sediment to be rapidly discharged from ice fragments. When collecting larger pieces of ice it was found that, in all cases, embedded sediment was rapidly washed out of the ice fragments largely within the melting of the first 10-20% of ice volume (Supp. Fig. 1). These ice fragments were specifically targeted to avoid ice with surface sediment layers and so this result cannot be explained by the loss of sediment frozen on the surface of ice. If this process was occurring at larger scales in nature it could further act to skew the deposition of

iceberg-borne particles towards inshore environments i.e. it would compound the inefficiencies in the delivery of sediment and associated nutrients to the offshore marine environment due to the rapid loss of basal ice layers. The mechanism of this process is unclear but it is not associated with ongoing cryoconite formation or similar associated processes due to albedo effects because the ice was stored in the dark.

**4.3 (Micro)nutrient fluxes to the ocean from icebergs**

By combining measured concentrations herein with estimates of the ice volume discharged from Greenland and Antarctica, annual flux estimates can be estimated for (micro)nutrients associated with icebergs (Table 1). For the macronutrients $NO_3^-$, $PO_4^{3-}$, and dSi, the uncertainty in these flux estimates remains large relative to the magnitude of the flux. This is an inherent result of the large fraction of ice with macronutrient concentrations close to the LOD, so would not be changed with further data collection. Iceberg-derived macronutrient fluxes are likely minor in terms of annual polar pelagic nutrient cycling (Table 1) and in most coastal environments will dilute, rather than enhance, ambient macronutrient concentrations. This is especially the case in Antarctic waters, where macronutrient concentrations are universally high (Boyer et al., 2018). The low macronutrient of ice also implies that physical effects associated with iceberg passage, mixing and any stratification resulting from meltwater are likely larger effects on annual macronutrient budgets for biota than the direct contribution of meltwater (Helly et al., 2011; Tarling et al., 2024). In regions where meltwater from icebergs accumulates in a thin surface layer, which is a phenomenon largely confined to Arctic fjords (e.g. Enderlin et al., 2016), low macronutrient concentrations may contribute to low primary production in near-surface layers. Although it should be noted that meltwater delivery is not confined to the surface (Moon et al., 2018) and, as noted, can drive the vertical entrainment of macronutrients within the water column.

| Nutrient | Greenland Ice Sheet annual discharge Mmol yr$^{-1}$ | Antarctic Ice Sheet annual discharge Mmol yr$^{-1}$ |
|---|---|---|
| NO$_3^-$ | 389 ± 345 (370) | 418 ± 796 (168) |
| PO$_4^{3-}$ | 18 ± 25 (14) | 76 ± 83 (58) |
| dSi | 212 ± 701 (27) | 476 ± 2187 (b/d) |
| dFe | 7.1 ± 15 (3.9) | 130 ± 472 (18) |
| dMn | 2.3 ± 6.0 (0.77) | 32 ± 191 (3.3) |

Table 1. Annual fluxes of nutrients associated with icebergs assuming calved ice volumes of 500 km$^3$ yr$^{-1}$
from Greenland and 1100 km$^3$ yr$^{-1}$ from Antarctica (Bamber et al., 2018; Rignot et al., 2013). Values
are mean ± standard deviation (median); 'b/d' represents a median sample below detection.

Delivery of total dissolvable Fe and Mn fluxes from icebergs to the ocean may be considerable (Table 1),
but, as these components are associated with heterogeneous particle-rich layers in ice, their delivery may
be skewed towards inshore waters where primary production is less limited by trace metal availability.
Dissolved Fe and Mn components are of more direct relevance to phytoplankton demands on the short-
term timescales associated with iceberg passage, due to the short residence time of particle associated
metal phases in the marine environment. Annual dFe and dMn fluxes also carry relatively large
uncertainties (Table 1) which reflects the wide range of concentrations present in ice. Although the crustal
abundance of Mn oxides is approximately 50× lower than that of Fe oxides (Rudnick and Gao, 2004),
dMn fluxes from Greenland and Antarctica are 32% and 25% of the corresponding dFe fluxes,
respectively (Table 1). Similar trends are evident in dFe and dMn concentrations within fjord
environments where trace metals from subglacial discharge and runoff enter the ocean (Forsch et al.,
2021; van Genuchten et al., 2022). The relatively-high concentrations of dMn compared to dFe likely
reflect the rapid scavenging of dFe close to source compared to more conservative behaviour of dMn over
short (hours to days) timescales (Kandel and Aguilar-Islas, 2021; Yang et al., 2022; Zhang et al., 2015).

A key finding throughout was that the macronutrient and micronutrient content of ice was relatively
similar between catchments and regions worldwide despite the contrasting geographic context of Arctic

and Antarctic ice calving fronts and notable differences in sediment loads between regions (Fig. 2). There was limited evidence of differences in ice nutrient concentrations between field campaigns returning to the same location (Nuup Kangerlua, southwest Greenland) in different seasons/years and similarly limited evidence of differences contrasting ice fragments collected offshore in Disko Bay (west Greenland), with ice fragments collected inshore close to marine-terminating glacier fronts (Fig. 5). Icebergs are inherently heterogenous due to the nature of englacial and basal sediment incorporation and loss processes. This heterogeneity combined with generally low nutrient concentrations, appears to mask and regional or catchment specific trends in macronutrient or micronutrient content related to changing bedrock composition (e.g. Halbach et al., 2019), calving dynamics (Smith et al., 2019), or photochemical processes (e.g. Kim et al., 2010).

Whilst further sampling would not reduce uncertainty in the estimated nutrient fluxes (Table 1), some specific caveats with our present work could be resolved in the future. Herein we have considered only $NO_x^-$ as a source of bioaccessible nitrogen, but considering the universally low concentrations present in icebergs, other N sources (e.g. DON- Dissolved Organic Nitrogen, and $NH_4$) may be relatively important. We hypothesized that a basal ice influence would be present in some ice fragments with high dSi alongside dFe and dMn, but conversely found very low dSi concentrations across all field locations. Future process studies might elucidate the mechanistic reasons why elevated dSi concentrations are not present alongside dFe and dMn concentrations in ice melt. Finally, sediment rich layers of large ice samples were observed to rapidly melt, potentially indicating that these layers are prone to disintegration. Such a mechanism could be an important regulator of sediment dispersion in the marine environment, potentially further skewing the delivery of iceberg rafted debris and nutrients towards coastal waters.

**5 Conclusions**

The dataset reported here covers ice fragments collected from a range of Arctic and Antarctic, polar and (sub)polar marine-terminating glaciers, and floating ice tongues. Throughout, icebergs are found to be only a minor source of macronutrients to the ocean with a large fraction of measurements close to, or

below the standard analytical detection limit. Icebergs do however deliver modest fluxes of dissolved Fe and Mn to the polar oceans, which are likely important ecologically- particularly in the Southern Ocean (Sedwick et al., 2000; Wu et al., 2019). The rapid dilution of meltwater close to icebergs, typically to concentrations <1% (Helly et al., 2011; Stephenson et al., 2011), means these trace metal inputs are challenging to constrain from in-situ pelagic observations (Lin et al., 2011), thus our measurements provide a first order constraint on iceberg-derived micronutrient fluxes into polar seas. The scavenged-type behaviour of dFe may explain why the dFe:dMn ratio in ice melt is considerably higher than expected from crustal abundances of Fe and Mn oxides, yet this also raises questions about how micronutrients sourced from icebergs behave immediately after release into the ocean. Dissolved Fe may be scavenged close to source limiting the spatial extent of Fe-fertilization from iceberg tracks, whereas, especially in the photic zone, dMn is more stable in seawater (Sunda et al., 1983). Thus icebergs may be an even more disproportionately important dMn source to biota than the dFe:dMn ratio in meltwater suggests.

**6 Data availability**

New data presented herein is available from SeaDataNet [ https://emodnet.ec.europa.eu/geonetwork/emodnet/api/records/ff3c625c-6a39-46ef-b329- 222040f85917, last accessed 20/08/2024]. Literature data was compiled from prior published values (De Baar et al., 1995; Campbell and Yeats, 1982; Forsch et al., 2021; Höfer et al., 2019; Hopwood et al., 2017, 2019; Lin et al., 2011; Loscher et al., 1997; Martin et al., 1990b). For convenience, a merged dataset is appended for data not previously compiled.

**7 Author contribution**

MH, DC, JH and EPA designed the study and acquired funding and resources. JK, DC, JD, JH, EA, TL, LM and MH conducted field work. EA, KZ and MH conducted laboratory analysis. JK, JH and MH conducted data analysis. JK and MH wrote the initial draft of the paper and all authors contributed to revision of the text.

## 8 Competing interests

The authors declare that they have no conflict of interest.

## 9 Acknowledgements

Tim Steffens (GEOMAR) is thanked for technical assistance with ICP-MS, André Mutzberg (GEOMAR) for macronutrient data, Stephan Krisch (formerly GEOMAR), Thomas Juul-Pedersen (GINR) and Case van Genuchten (GEUS) for assistance with sampling. The captain and crew of RV Sanna are thanked for field support. Antarctic sampling was possible through FONDAP-IDEAL 15150003 and FONDECYT-Regular 1211338 (awarded to JH). MH received support from the DFG (HO 6321/1-1), the GLACE project organised by the Swiss Polar Institute and supported by the Swiss Polar Foundation, NSFC project 42150610482 and the European Union H2020 research and innovation programme under grant agreement n° 824077. LM was funded by research programme VENI with project number 016.Veni.192.150 financed by the Dutch Research Council (NWO). JD was sponsored by a scholarship from the Instituto Antártico Chileno (INACH), Correos de Chile, and the Fuerza Aérea de Chile (FACH). Ship time and work in Nuup Kangerlua was conducted in collaboration with MarineBasis-Nuuk, part of the Greenland Ecosystem Monitoring project (GEM). We gratefully acknowledge logistics and funding contributions from the Danish Centre for Marine Research (DCH), Greenland Institute of Natural Resources, Novo Nordic Foundation (NNF17SH0028142) and INACH.

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
