# Peer review of "The macronutrient and micronutrient (iron and manganese) content"

_EGUsphere, 2023_

## Author Response (AR1)

**We would like to thank the Editor and two reviewers for their time and constructive comments on the manuscript. In drafting a revised text we have tried to accommodate the comments into a new structure and also clarified some small ambiguities raised (in many cases the new structure makes the minor comments by line obsolete). Comments on the text are in italics; replies are annotated below each comment.**

*The paper is well written, the majority of the diagrams are clear and the data tables in the supplementary information extremely valuable and easily formatted. I would have preferred maps rather than nice photos to aid interpretation, but I will leave this to the judgement of the reviewers. I also note that some of the chemical notation was not quite correct (nitrate, nitrite and phosphate ion abbreviations should include valency), but I am happy to correct this at the next stage of the review process.*

**R: Several comments suggested a chart instead of photographs, noting the general literature often muddles sea ice and glacial ice we opted to keep photos and introduce a chart to the main text. Ion abbreviations are now strictly correct ($NO_3^-$, $NO_2^-$, $NO_X^-$, $PO_4^{3-}$ and dSi)**

Review

*The authors have compiled a unique dataset of 589 iceberg samples, including 367 new samples. The authors present a suite of macro and micronutrient concentrations for these samples. The manuscript represents progress beyond current data availability and some new insight into the relationship between nutrient concentration and particulate load/ice melt rate.*

*The purpose of the work clearly articulated. The possible sources of nutrients in the glacier should be more clearly listed and explained in the introduction. The methodology needs more indepth discussion of models used to test nutrient source. Interpretation is underpinned by data presented. The reviewer is not an expert on the models and cannot comment on their validity – but would appreciate more detail on how the modelling methods can achieve the aim (alongside previous examples of their use). Results are presented in a robust way. Possible bias as a result of possible sea ice sampling is discussed in detail and lab methods are implemented to reduced possibility of sea water supply of nutrients.*

**R: We struggled in the original draft to focus on 'possible sources of nutrients in icebergs' without moving into topics which may not be clearly related and don't keep a tight focus in the text (another reviewer suggests we still should consider a tighter focus than in the original text). Biological processes on glacier surfaces (e.g., ice algae) for example are alluded to in several comments, as are basal processes associated with the unique chemistry of the layers of ice at the glacier bed. But it is conceptually challenging to state that these processes are related to iceberg nutrient content because the geological consensus is that these**

'peripheral' ice layers are lost before/during ice calving so they don't affect the composition of icebergs at sea. This somewhat verified in our results as we find no evidence of any high concentrations of dSi in 'basal' ice layers in our entire dataset. Atmospheric processes, which influenced the composition of precipitation that ultimately became glacier ice and calved into the ocean, are similarly the main 'original' influence on ice composition, but these processes are a long way detached from the focus of this work. It is therefore conceptually difficult to draw a line between topics which are vaguely and definitively within scope of a 'source-to-sink argument' for icebergs. The 'source-to-sink' concept itself as framed in some comments is also potentially flawed if we think about formation of marine ice and scouring which interrupt the notion that sources are associated with glacial processes and sinks are associated with melting. In the revised introduction we now convey this uncertainty by phrasing the issues as open questions which research (herein and in the future) might help resolve.

The 'models' referred to in this review are statistical tests and not models. We have added a few more lines better explaining the statistical tests and their implications. A key observation in our dataset is that there aren't many differences between different subsets of data which suggests, in simple terms, that iceberg nutrient concentrations can largely be assumed to be homogenous in terms of at least macronutrients. To our knowledge, there are no large-scale similar efforts to explore regional differences in ice chemistry, other than sea ice studies which have found critical differences between ice layers sorted by depth. Yet these studies are generally using small datasets and concern individual profiles and parameters rather than statistically tests covering multiple parameters and locations.

*Significance (impact):*

*The manuscript provides a step towards understanding nutrient supply to the ocean by icebergs. The strength of the manuscript is the large ice dataset and multiple nutrients considered. No substantial conclusion is reached. The authors conclude that micronutrient content from icebergs (e.g., Fe and Mn) is similar geographically. However, to provide weight to this argument requires i) more detail presentation of the tools used to reach this conclusion e.g., how the models work and previous use of models.*

R: There were no models in the original (or new) text, rather statistical tests. We have better explained this in the methods to improve clarity.

*In addition, it is limiting to use just iceberg concentration data to infer glacial source processes that drive the initial concentration changes. For this mineralogical characterisation, nutrient – organic carbon associations characterisation, isotope geochemistry to detect sediment source.*

R: It should be noted that almost all of the samples would be below detection for isotopic analysis, so any isotopic analysis would be heavily skewed towards the

few samples with concentrations above detection limits. We allude to this in the comments added in response to Reviewer 2 concerning how data is treated (the large percentage of data below detection for phosphate and silica, for example, would limit isotopic analysis and skew interpretation). The same applies for the comment regarding mineral characterization and organic carbon. Any such analysis would be insightful, but proper consideration would have to be given to the skewed distribution of samples for which this analysis was possible. As noted the framing of the study in the context of 'glacial source processes' may not be correct as the geological and oceanographic consensus is that most basal ice layers are lost prior to icebergs calving and thus atmospheric source terms are the clearest source in the data.

*Suggestion also to deconvolve ice nutrient concentration into: what processes drive initial concentration in the glacier; what processes could change nutrient concentration when iceberg is calved, what processes could change nutrient concentration when the iceberg is transported (e.g., cryoconite and rate of melting). The distinction between these points is discussed in the paper, but could be presented more logically so that the reader has a logical picture of the sequential reasons why nutrients to be of a certain concentration in the ice. The paper would have more impact if presented as 'what can possibly affect the iceberg concentrations from source to sampling'. If the reasons are weighted towards iceberg melt/ice berg geochemical processes then this would put more weight on insitu and reduce weight on geographical source being governing factor. By framing in a 'source to sink' approach it links with the papers motivation to understand if there are geographical distinctions in nutrient source.*

R: In the revised text we have adopted a 'source-to-sink' framework in an effort to reframe the introduction/discussion following reviewer comments which does, we agree, improve the structure and hopefully help the reader follow literature from multiple fields. Yet, as noted, we cannot give much insight into 'sources' because many of the processes that affect glacier ice nutrient concentrations, particularly higher concentrations, occur at the glacier or iceberg periphery with most of this 'peripheral' ice lost before or during calving. As the reviewer comments, this does lead towards a clearer summary/conclusion that the melting of icebergs in the ocean is a major influence on the measured concentrations rather than source specific features which are not preserved long after calving. Not much insight can be gained in our work presented, for example, into basal ice processes occurring at glacier-beds along large ice shelves because basal ice is completely melted below ice shelves before ice is calved into the ocean.

*What is meant by micronutrient signature in the abstract?*

R: We meant concentrations and ratios of nutrients to each other which can be insightful concerning the origin of nutrient sources, this has been changed throughout to improve clarity.

*Please explain in more detail why you would have expected a geographical difference in nutrient concentration – is this the result of variability in sediment supply and geochemical processes in ice sheet glaciers globally? If so, please state this evidence.*

**R: The source-to-sink perspective now better clarifies this as we discuss the potential implications of different contexts. As noted above, the concept of geochemical processes at glacier beds affecting nutrient loads in icebergs is perhaps misconstrued as the consensus (largely verified by our results) is that sediment from basal ice along large ice shelves and tongues that calve icebergs is lost prior to calving, so is not present in our dataset.**

*Unclear what is meant between dissolved and total dissolved in these two sentences: Total dissolvable Fe and Mn retained a strong relationship with sediment load (both R2 30 = 0.43) whereas weaker relationships were observed for dFe, dMn and dSi.*

**R: 'Total dissolvable' and 'dissolved' are now further clarified and defined.**

*Suggest to better link the 'sediment load loss' discussion - to one part of the continuum from sediment nutrient supply in the glacier to iceberg sampling. This continuum could be presented in the introduction and again at the start of the discussion.*

**R: As per the structural comments, a 'source-to-sink' perspective is now used to improve the structure of the introduction and discussion. We cannot fully constrain the effect of 'sediment load loss' because the largest effects of this process have probably already occurred prior to iceberg calving underneath floating ice tongues and shelves which we do not study herein.**

*A very comprehensive introduction. One point: please clarify what is meant by sediment supply of micronutrients. Does a portion of the sediment supply refer to the supraglacial sediment deposits? Do these deposits interact with supraglacial meltwater and transport micronutrients to englacial systems? Also are cryoconite holes another possible source of micronutrients? Suggest that you provide a clear summary list of possible sources of micronutrients, supported by literature.*

**R: As noted in the revised introduction, sediment in ice can have many origins, we have included additional details to the introduction to clarify this. We also note that the geological consensus is that most sediment from larger ice shelves is lost prior to ice calving, so many of these processes are not so relevant to samples collected from icebergs at sea.**

*In the introduction, please make a clearer link between 'sediment interaction' and tidewater glaciers. In the introduction, what is meant by sediment-rich peripheral layers?*

**R: These points are now related and clarified as part of the 'source-to-sink narrative'. General consensus in the literature is that the highest sediment loads can be found in some specific contexts: basal ice associated with basal**

weathering processes, or surface ice associated with land-slides onto ice or cryoconite. These high sediment loads can end up embedded in the glacier's interior due to ice movement, but are generally clearly identifiable as distinct layers in glacier termini and in freshly-calved icebergs. We refer to these collectively as 'peripheral layers' because in the context of a calving iceberg these layers are vulnerable to rapid loss and in the case of basal layers may not be present at all in our dataset.

*Line 154: what is meant by 'total dissolvable' compared to dissolvable? Does this include a portion of particulate?*

R: Total dissolvable is defined in the methods, it means all metals which can be dissolved in HCl at pH2 over the storage period of the samples (>6 months), so yes this includes a portion of particulate metals defined as the maximum portion which could physically be leached into solution without further weathering. ('Total' would be obtained from a concentrated $HNO_3$/HF digest which would then include refractory lithogenic phases which in most contexts would be deposited in marine sediments without having undergone any processing in natural waters during the 'source-to-sink narrative'). To further clarify this, we added an extra sentence to the methods.

*Please provide more information on how PERMANOVA works? Are there examples of its use for similar questions in other studies?*

R: PERMANOVA (permutational analysis of variance) is a standard statistical test, a type of ANOVA (analysis of variance), which is standard in studies comparing the centroids of different groups in multivariate analysis (several dependent variables instead of just one like in ANOVA) with large datasets. Because iceberg (and sea ice) studies are often small and focus just on one variable (e. Fe concentration), they have to date- as far as we are aware, generally relied on qualitative comparisons of concentrations rather than statistical analyses. PERMANOVA analyses are more appropriate for multivariate larger datasets, as ours. This is a common statistical test to assess differences among groupings using multivariate datasets such as studies looking into potential differences in community composition among sampling sites. For example, a PERMANOVA would be a common choice of test for a dataset with species abundance data for many species across multiple field sites.

*Please provide more information on how MDS works?*

R: As above, this is a standard unconstrained ordination statistical test. To improve clarity, we have added a sentence about each test in the methods section and also added a few sentences in the figure legend to explain to a reader how to interpret a nMDS ordination plot.

*In Figure 1 caption, please provide information on what is meant by MDS1 and MDS2. Please explain what an nMDS ordination analysis is.*

**R: As above, we added a line in the methods to improve clarity. In simple terms, it's a statistical test to see whether or not there is grouping of the samples into specific regions. We are testing whether the region where a sample was collected is an important factor affecting (micro)nutrient distributions. A few sentences have now been added to the legend too. The advantage of a PERMANOVA is that it considers all of the nutrients' simultaneously, rather than simple comparisons of each nutrient in each location one at a time.**

*Line 273 – 274: 'despite the potential for dSi to be released from sedimentary phases via similar mechanisms to Fe and Mn, neither trace metal correlated well with dSi.' – what do you mean by 'release by sedimentary phases via similar mechanisms'? Suggestion to include sentences on these mechanisms in the introduction. Please make the possible sources and mechanisms for sediment to dissolved phase transfer clear for all nutrients in the introduction.*

*Very good section on inshore to offshore concentrations in ice and possible contribution of elements from sea ice.*

**R: Some brief explanations can be added here but there aren't actually many studies specifically commenting on the specific underlying mechanisms for the dissolution of particulate Fe/Mn/Si into dissolved phases from ice-bore sediment. Most studies concern dissolution/precipitation dynamics in aqueous solution in the context of seawater or freshwater and solid state ice chemistry may be fundamentally different. What we meant to convey here was that these elements (Fe/Mn/Si) have high concentrations in lithogenic borne particles and that in other environmental contexts Fe/Mn/Si can show high dissolved concentrations in aquatic environments from weathering processes (as per runoff at our sample sites)- whereas P and N species generally do not. In hindsight, this was not well explained and we have edited accordingly to improve this text.**

*Line 426 – 429: 'This is consistent with the expectation that englacial sediment drives a direct enrichment in TdFe and TdMn, which increase proportionately with sediment load, whereas the enrichment of dFe, dMn and dSi is more variable and depends on the specific conditions that sediment and ice experience between englacial sediment incorporation and sample collection.' We are not introduced to the expectation that englacial sediment drives enrichment of Fe and Mn. Suggest to explain the possible mechanisms of sediment micronutrient enrichment in the introduction and support with literature.*

**R: As per earlier suggestions, the introduction and discussion are now re-structured to provide a 'source-to-sink' perspective so this hypothesized process is explained in the introduction. We have introduced the difference between "total dissolvable" and "dissolved" in the methods section as there were several**

technical queries about this in comments which are not easily answered in a sentence without introducing multiple phrases about 'particulate' or 'labile particulate' elements, which might become confusing to a reader.

*Suggestion to graphically show the relationship between melting rate and sediment release rate.*
*Citation: https://doi.org/10.5194/egusphere-2023-2991-RC1*

**R: We had already plotted this in the original Supplement (Supp. Fig. 1) which now also shows the fit formula. We prefer to keep this graph in the supplement because the experiments used to deduce the rates were quite basic and so the rates were probably not scalable to in situ conditions. Accordingly, it is more of an interesting hypothesis for further work/a caveat when thinking about this work, rather than a main feature of our findings.**

*Review Krause et*
*al.* *07/06/2024*

*Krause et al. accumulated an impressively large and unique dataset of nearly 370 (not 600 as expected from the abstract) ice samples from both hemispheres. These were then analysed for macronutrients, several micronutrients of which mostly Fe and Mn (dissolved and total dissolvable in some samples) concentrations were shown. These were compared to the sediment load of the sample. These concentrations were methodically statistically analysed and for the td fraction a correlation was found in contrast to the dissolved fraction and local processes seemed to be more important than region, climate or geology.*

**R: The number of samples stated in the abstract is correct, the reviewer was perhaps referring to a subset of the data (most of the data was already online as some subsets from specific cruises were uploaded as open-access datasets when the respective cruises were complete – we did state this in the 'Data availability' section but have added an additional sentence for clarity in the methods, put a subheading in the methods to clearly show the section on data compilation, and added a supplementary table).**

*The English in this ms is fine and the dataset is a wonderful collection which contains a lot of work and the authors must have put a lot of thought into the statistical methods. Also, I enjoyed reading the discussion. Now to my concerns in regard to the ms though and I hope I don't come across too harsh as I know how much work and brain power and time must have gone into this ms before submission - especially for the first author. Therefore, I have put a fair bit of my time into it to improve it as I think it could certainly be improved with more structure and focus and possibly a clear message. And the ms needs to be narrowed down. There is a lot of information about a variety of polar and marine topics that the reader can get lost which I don't think helps and I think by restructuring the ms, it will be more read and especially cited. Sometimes less can be more even though it is very hard to cross out your own sections that have been written*

*with a lot of detail and thought, but you might be able to reuse them for another ms or so in the future.*

**R: We thank the reviewer for their comments and have no objections to shortening or better focusing the text, although this is not so easy to align with comments making requests for more information in several sections. In considering the balance between breadth and depth we have adopted the 'source-to-sink' suggestion from Reviewer 1 to help structure the Introduction/Discussion, but we have also removed some discussion and spent more of the word count introducing the basics (as per the next few comments about helping the reader and the issue Reviewer 1 raised with statistical tests which were misinterpreted as models) and less raising 'side-issues' which detract from the main story. Thus, we have cut discussion of trace elements other than Fe and Mn for brevity and to keep a tighter focus.**

*So, my suggestion would be to restructure the ms with potentially a new title (maybe rather the findings? But I think this question will be answered at the end of the revision), clear subheadings and structured paragraphs and leading the reader with your figures and first of all a global map in which you please show your sampling sites and a simplified sampling site name or starting with a table in the intro in which you name and classify the sampling sites (Hemisphere, region, climate, geology, which parameters were sampled, how many samples taken,…). At the moment I find very confusing to understand which of the different sampling sites each section is referring to (I know the person who sampled is usually very familiar with the names and sites, but a reader might not be and in this case not even be sure sometimes which hemisphere you are referring to and this needs to be clarified, please, it might be helpful to put an S or N or something into brackets) and the ms is jumping between topics and regions - the Arctic and the Antarctic. Both marine systems need to be better introduced, differences, similarities and the importance of ice, ice formation, nutrient input and trace metals in each. Then you can refer to these better in the discussion and might get a clearer message/conclusion in the end as well if all these are clearly laid out.*

**R: The manuscript now has more text which introduces the basic descriptions of the data requested in the first part of the results section, which will help the reader. The simplest key findings are that icebergs have a regionally invariable, low  macronutrient concentration. We have slightly edited the title. As requested, (also by the Editor), a new figure shows the origin of samples (new Fig. 1) and a Table (in the supplement) lists the exact sample collection details.**

**A key 'problem' we faced with the data is that some subsets are better than others for specific questions. For example, only one subset of data (in Disko Bay) collects ice both inshore and offshore at the same location, thus making an interesting case study for asking whether or not nutrient concentrations change as ice moves offshore. To try and address this issue we have better introduced the basic presentation of the data at the start of the results section and then in**

**each sub-section of the discussion explained the rationale behind specific exercises.**

*Also, it needs to be clarified if these 600 samples and their corresponding data and their publication are completely novel (this is what I thought in the beginning, but whilst reading the ms, I wasn't sure any longer) or whether this paper is rather a collection of all the ice samples published until now or whether it is a mixture of both? Not clear to me, sorry. and that this paper is rather their statistical evaluation? No matter what, it is necessary to clarify this and somehow create an overview table, I know some information is in the tables in the SI, but still I think this information and an overview table with the sampling sites (the above-named regions, classifications, geography, sample types taken, year, season, number of samples,… and please feel free to add) is necessary in the main paper. As you should be able to understand the main messages of a paper when skipping through it and reading the title, possibly the abstract and checking out the figures and their figure captions and the possibly the discussion, but certainly the conclusions. In contrast possibly you could think about the long and detailed methods section and whether parts of this section could go into the SI instead?*

**R: As suggested, a Table is added about the basic origins of all data (information concerning new samples and data re-used from the literature was already included in the data and method sections of the original text, further details are now added in the Supplement). Almost all of the samples are 'novel' in the sense they are measurements which have not been published in manuscripts, most of the samples (all except 14) were measured in our laboratories but several sub-datasets have been released as part of data products from specific cruises according to open access mandates. Also, in some cases Fe data from samples has been discussed but not the corresponding nutrient or Mn data from the same samples. 367 of the samples are completely new (new analyses, no prior literature), 14 come from literature by other authors, 208 have been partially (mainly for Fe) published in prior work- the analyses for these samples (both that already published and new to this work) came from our laboratories using the same protocols. The methods in the revised version are longer due to the requests for further details in the review. We prefer to keep these in the main text rather than splitting into a supplement.**

*I think a pretty good example and paper that I was thinking about which might help in terms of structure is D. Lannuzel et al., Elementa Sci Anth, 2016*

*Iron in sea ice: Review and new insightsIron in sea ice: Review and new insights*

*When I was looking for an example to give to you, I came across this one (and no, I am not an author of it 😄) and it actually has a clear message title, two different introductory sentences for the abstract and the intro, setting the scene for each, but differently. Then Table 1 to give the reader an overview of the sampling sites shown in figure 1 (you could do a Southern hemisphere and a Northern hemisphere global map as panel A*

*and B for example) and then Table 2 for the parameters used (not all of them, even if you measured a lot more, might be necessary for the story)*

**R: We have added an overview figure and table as suggested to clearly annotate where samples came from. The paper cited has a nice structure, but it is a review paper and thus 'easier' to have a non-conventional structure. Here we have to make a lot of effort to summarize new results, methods and statistical analyses which obviously have less flexible formats and we think are needed to help address the points raised by Reviewer 1. In considering the perspectives of both reviewers, we have re-written the introduction to address the major comments about the text. It should also be noted there is far less literature specifically on the biogeochemistry of icebergs than sea ice, so again it is challenging to keep a tight focus on icebergs whilst bringing in new ideas about source-to-sink narratives that concern literature from other fields as per Reviewer 1. Beyond our own work, there are only about 14 prior samples of ice measured in the literature (plus a few more in papers that do not provide the underlying data), so it would be extremely challenging to follow the sea- ice review structure as suggested.**

*Details:*

*L1: The title led me to expect a ratio or several (signature) of the measured nutrients and especially Fe and Mn and how to use them to identify their sources, the prominence of the ice and to learn more about their input into seawater, possibly elevating the (co-)limitation in the SO and how it could be compared to the Earth/sediment ratios. Potentially add something like "fixed" or "static" to show in the title that the signature is in a small range? At the same time you had some samples which you excluded from the stats due to their values…. As mentioned above whether the title is completely fitting will be answered after a detail revision. L24 main message in regards to macronutrients- are these necessary in the title?*

**R: We have revised the title and removed 'signature' throughout.**

*L15: is the first sentence in line with the title? It should be setting up what the reader expects of this ms. Potentially rather an introductory sentence for the introduction.*

**R: Edited to be more in-line with the title.**

*L25-30 lots of numbers, are both medians and means necessary, but there are no errors, ranges, which might be easier to grasp? some with stats, R2, but missing the p-value in L28. Please try to present this in a more uniform way. Jumping between dissolved and tD elements.*

**R: Writing a general overview, we have to consider that many of the samples were below detection which influences the most appropriate choice of stats whilst staying in the word count for an abstract (we have now better explained this in the text). Both $R^2$ and p values are now quoted, while for others we stick to the**

median. Dissolved and total dissolved fractions of dissolved trace metals are both standard measures which are more useful when measured together, so we prefer to state both.

*L27 suddenly total dissolvable, please try to lead over from the dissolved phase, maybe with a "in contrast to…"*

**R: This would lead to two sentences back-to-back with 'in contrast', we prefer the sentences as it was.**

*L25 and throughout the text: chemically I would prefer the writing $NO_3^-$ for nitrate and $NO_2^-$ for nitrite as $NO_3$ would be rather a nitrate radical and $NO_2$ laughing gas (nitrogen dioxide).*

**R: Done. Edited throughout.**

*L32 please add a number behind the Arctic to make it comparable.*

**R: Added.**

*L33 what does this mean losing their sediment load? Message of this finding? Very interesting and curious*

**R: It means the sediment load is lost, i.e., icebergs likely do not have a constant sediment load because sediment is not distributed homogenously, it declines sharply after calving.**

*L35 correct to $PO_4^{3-}$*

**R: Done. Edited throughout**

*L35 d or td?*

**R: Throughout the element e.g. "Fe" refers to all phases of that element, we have used dFe and TdFe only when specifically referring to the respective measurements.**

*L42 nutrient availability? Not clear what is limiting and where, potentially easier to explain if the regions are introduced first?*

**R: There isn't much certainty concerning the exact identity of what limits primary production in each region, other than broad statements about Arctic systems likely being N/light limited and Southern Ocean systems being Fe/Mn/light limited. In this specific sentence, we are referring to mixing which would increase vertical fluxes of all nutrients simultaneously so it doesn't actually matter what specific nutrient was potentially limiting phytoplankton growth.**

*L50 Reg needed after ice*

**R: The introduction is re-written, this comment may be obsolete.**

*L52 Ref necessary*

**R: Added.**

*L70 what are the biogeochemical consequences?*

**R: These are listed throughout this paragraph in the following sentences.**

*L78 new topic so potentially a new paragraph?*

**R: We think this is the same paragraph.**

*L87 does it matter in which region, hemisphere we are here? Sorry, lost me*

**R: No, these are universal processes. We have edited line 90 for clarity.**

*L113: if the nutrient signature of icebergs hast been published, maybe give some numbers/ratios, introduce them for the different regions, settings – if easier with a table?*

**R: The new results section starts with a basic distribution and data analysis which hopefully addresses this comment.**

*L116 to the polar ocean*

**R: We are referring to work which derives global fluxes and thus in some cases these are not polar.**

*L128 if this is all about the signature an how to use it, how it varies, the signature needs to be introduced better and what it could be used for? Or do you simply mean with signature that the introduces a number of macro- and especially micronutrients?*

**R: We have removed the term 'signature' from the revised text.**

*L135 and throughout the intro, you are using d, td and p fractions and I think it is necessary to introduce them very briefly*

**R: Done. Edited sentence to contain a general introduction to the concept.**

*L128 and 140 you have two clearly formulated hypotheses. Maybe number them and use these to build the ms clearly around these? And please try to answer them and it might help with the golden thread of this story.*

**R: Edited. We now finish with two clear hypotheses which are returned to in the discussion.**

*L149: add subheadings, please. Start with sampling and sampling sites, sampling methods, sediment load experiment, analysis (nutrient and TM)….*

**R: Done. Subheadings added.**

*L151 why were the samples directly thawed and not kept frozen?*

**R: Samples have to be melted to be analyzed for concentrations. Shipping ice samples from remote locations is also extremely challenging and simply would not be possible in most of the sampled cases, especially not with a guarantee that samples had not been subject to some degree of melt and re-freezing en route which would maybe change the measured concentrations.**

*L153: pre-cleaned filters and syringes for TM analysis? GEOTRACES protocol?*

**R: Edited, (yes filters and syringes were pre-rinsed with 1 M HCl and then with a small quantity of sample to pre-rinse). 'GEOTRACES protocol' is vague as the GEOTRACES cookbook does not give one specific protocol, it reports recommendations from multiple laboratories, hence we wrote out the exact procedure used already for full clarity and transparency.**

*L155: new paragraph, potentially this needs a map/figure for Disko Bay or potentially a table to show what for and where these samples were taken, as mentioned above – or move L151-155 as sampling below all the sampling sites, to sort this in a better way.*

**R: A new table and figure are added to show the origin of all samples (new and literature).**

*L167 as written above this would better lead over to this section above sampling. Not clear to me were the wet sediment-sub-samples come from as they haven't been mentioned before. Possibly change the order og the sentences here as well to clarify this?*

**R: Done. Order changed as suggested.**

*Very interesting experiment, not sure why the sediment was decanted each time? And which in which phases what (d and td was sampled in the seawater or in the mixed seawater thawing ice samples, which parameters were taken? And how?) potentially a table or figure could help to explain these experiments? It feels like there is sooo much information and interesting samples, sites and a sediment load experiment in this ms, but they can get lost if not clarified or possibly split into two ms that could be accompanying each other?*

**R: We are not inclined to make multiple manuscripts where one would possibly suffice considering the time and effort required to review them. In this experiment the sediment had to be decanted to see if there was any skew in when the sediment was released (i.e., if it all came out straight away or if there was a slow and steady release of sediment).**

*Also here L178: better to have a complete paragraph about one analysis, then the next, very convoluted.*

**R:    Methods re-written as suggested.**

*L179 and 181 which grade?*

**R:    Reagent grade (added for clarity).**

*L182 why where the samples standing upright for more than six months? Because otherwise and addition of HF would have been necessary? Please add a sentence and a Reference*

**R: Samples are stored upright to prevent them leaking or leaching the lids from LDPE bottles as the caps are not LDPE (usually PP). There's no specific reason for 6 months, it takes a long time to get samples shipped from remote places and scheduled for analysis. Especially with total dissolvable analyses it's better to standardize the leaching time in case it does make any difference – there are some technical papers on this which are a bit out of scope, we do not want to get into a lengthy discussion about leaching times to define 'Td' as there is no standard protocol for 'Td' throughout the literature other than the general statement that 'Td' is unfiltered and acidified.**

*L183 sample volume?*

**R: We have given vial volumes.**

*Where all the TM samples (p, d, td) measured in the same way? Usually different procedures and preconcentration-7dilution etc processes necessary, not clear at all. Restructure and clarify please. What about blanks, repeat seawater or or samples and errors and detection limits? Methods and references to these?*

**R: We have now added more detail about the dilution protocol (because of the huge range of metal concentrations encountered we initially ran samples with a moderate dilution factor, then re-ran those which had low Fe or Mn concentrations). The blanks and detection limits obviously vary over the multi-year period over which all these samples were analysed and cannot be presented without tables of data to show each individual batch. A summary is provided with the highest/lowest batches quoted showing they are acceptable compared to the sample concentrations.**

*L189 first time the additional TM Ni, Cu and Co were mentioned (UV? Of the samples?)*

**R: "Exactly as per Rapp et al., (2017)", so yes UV treated. This section has been removed (trace metals other than Fe and Mn are now not discussed).**

*L198: varied by how much?*

**R:     An extra sentence has been added regarding variation.**

*L200 in my opinion this needs to be clarified before as! As you usually write about your 367 samples, their analysis etc and then use the other compiled data in the discussion to compared and, yes, sure for stats and modelling, but potentially think about separating this and the question is whether it might clarify your story…. Were all these samples from the Arctic? Is the Antarctic necessary or only then in the discussion? This way the title might be more adequate as statistical analysis of…*

**R: The new table and figure address this. We are usually writing about all of the samples in the statistical analysis, and as such it does not make sense to separate 'new' and 'already published' data as in almost all cases these come from the same laboratories and same regions. Only 14 of the total samples do not come from our laboratory.**

*L208 it is necessary in this section of stats to explain with a sentence why which was chosen and what it is used for*

**R: We already stated this in lines (originally lines 216-218).**

*L220 if removed these samples need to be discussed and if the TM ratio were still the same?*

**R: Line 222 already discussed the effect. Including outliers in any dataset skews basic statistics, hence why they are removed. As far as we are aware it is not normal practice to explicitly discuss the effects of outliers in general unless there is a specific reason to do so, oceanographic datasets are always cleaned prior to analysis. Data cleaning procedures (Quality Control protocols) are applied using standard protocols before data analysis starts to avoid any such effects.**

*L224 eplain this in 2-3 sentences*

**R: The next 3 lines do this.**

*L226 below DL, has this been mentioned in the methods? If the treatment mad a difference, how did you treat them and what happened, what were the different results and please add Ref*

**R: Yes, the LOD values were already given in the methods. We are not sure what the reviewer wants a reference to, it is basic knowledge that when a dataset contains a large amount of data below detection it makes some difference whether the data is assigned a value of '0', a random value 0<x<LOD, or a specific value of LOD or 0.5 LOD, or another nominal value. Changing this obviously makes some small differences to stats, we have specified exactly what we did to handle this issue herein, we do not think it is necessary to give multiple alternative approaches given that there are a large number of possibilities. There is no standardized way of handling a dataset where values are <LOD and very extensive text would be required to thoroughly test all possible treatments of the data which would turn the manuscript into a statistical assessment. In a few sentences we have explained these issues and tried to show the 'largest' possible difference by deriving basic statistics in the case where data <LOD is ignored (see new section 3.1).**

*L241:td has been introduced before*

**R: We have edited this.**

*L244 a range or rather a concentration with plus and minus might be easier to grasp?*

**R: We respectfully disagree here, as the ranges are huge and the standard deviations are relatively large, so we do not think these metrics are useful.**

*L249 add number to show this, overall not clear where all this is heading to. A lot of information but what about the order and common thread?*

**R: We don't know what additional number is being requested here, we think the lines are clear as written.**

*L254 at catchment level- this is what I meant rather with the table and sorting to show which subset you are referring to. Possibly add this with a colour code or a symbol or so in your figures*

**R: The new table and figure added hopefully now better display the data.**

*L259 I don't think that every reader will understand this figure with a MDS versus the other. In cases of concentrations okay, but these haven't been presented until here at all and you are jumping straight into this which is not leading the reader. Not clear what can be read from this. This needs to be better explained in the text. Also it is necessary to clarify which data is novel and which from the literature (then refer to them correctly, please) and honestly I found the tables and figure in the SI more enlightening and interesting than this one. Maybe with more context…*

**R: MDS is a relatively standard statistical technique. We have added the extra figure and table as requested to better introduce the samples. All the data plotted**

without citations is novel (note that some data subsets were posted open access as part of data products from their respective cruise/field campaigns, but they are all 'our' data and have not been published elsewhere so are all novel, only 14 values are literature values not generated by these authors which was noted in the prior text and is hopefully now additionally clarified in the Supplement).

*L266 if it would skew the relationship, please explain how and why before simply excluding these samples*

R: In any linear plot extremely high (or low) values skew the gradient, we do not think this needs explaining in detail, as it is common to any linear regression. It is standard protocol to exclude outliers from any dataset before interpreting the clean dataset. Outlier inclusion and effects should not be discussed in depth, as it does not add much value to the scientific interpretation.

*L269, 270 rather discussion*

*L274 add Ref.*

R: We are referring to our own work/values, thus no reference is required. As this is just a comparison of our ratios to literature ratios we prefer to leave it in the results section.

*L285 the grey area needs to be explained and mentioned. Add the relationship in numbers of the correlation, please. I would also keep the symbols and colours the same in all the plots. And why not start at 0 with the axis and finish accordingly?*

R:   Grey area is now defined (95% confidence). We note that if the plots are plotted as 0-0 with no offset, the bottom left corner becomes unclear and cluttered as so many points cluster at the bottom right.

*Also if all the units per L-1 then the question is, why you us M and not mol L-1 then it would be all more uniform and more obvious?*

R: It is relatively standard in the field to use "µM". All of the new results are meltwater as stated so there is no difference between $L^{-1}$ and $kg^{-1}$.

*L291 which subset now and why was it chosen?*

R: The next sentences explain this in the original version, there is a subset of data within the whole dataset which sits on the 1:1 relationship, we didn't 'choose' this data and the sentences already explain this distribution.

*L297 if only used 1 per site or a median or mean for each site/category what would happen then?*

**R: We already stated that these data account for a small fraction of the dataset "14% of the sub-dataset where all macronutrient concentrations were detectable" so they are not reflected in mean or median values for each site/category.**

*L301 and 307discussion mixed in*

**R: We considered moving a section on 'Data quality' or similar to the discussion but prefer to deal with issues concerning quality control conclusively in the results section so the discussion section is cleaner and stays on topic.**

*L313 add Ref.*

**R: The references in the next sentence are the ones which cover this point.**

*L320 not sure why these samples (additional not new ones) where chosen for this compilation? Are these all the samples available? Or because they were from the same group of authors? Then even more a good discussion and statistical evaluation with other samples would be important to put this dataset into perspective.*

**R: (Almost all of the iceberg data is from the same authors, only 14 of the samples available are not from our work). These specific samples were mentioned here because they are from times where we have in situ pelagic data and iceberg data at the same time.**

*L330 add Ref.*

**R: These were already included in lines 315–320.**

*L331 unprecise: a few samples were collected. n=?*

**R: The next sentence gives the two values. We have edited this to 'two' for improved clarity.**

*L333 range, unclear*

**R: When giving two values, the range is the same as the span of numbers.**

*L338-342 good and interesting but rather discussion? add Ref.*

**R: We prefer to keep the details about address sample bias in the results section because this helps keep the discussion cleaner. We do not think we need a reference to state that glacial ice has zero salinity, as this is the case by definition.**

*L344 add Ref.*

**R:We    added a reference.**

*L345 if this is a critical difference what does it mean for the data, any idea about the differences?*

**R: We can only speculate and this would require further analysis/discussion which we do not have space for.**

*L350 nice pictures, but still no map and not clear where exactly these were taken. Not exact enough. Global or regional importance? Not clear enough and getting lost here what the importance of the paragraph for Disko Bay is just trying to follow this in the text. Therefore new figures are necessary for this.*

**R: We have added a new overview as suggested in the form of a figure and table.**

*L413 discussion*

*L421 discussion*

**R: This was edited**

*L446 Supp Fig. 1 very interesting, I would prefer this in the ms, why SI?*

**R: The sediment story is a bit of a side-story which doesn't fit in the main narrative that the reviewer has encouraged us to stick to. Therefore, we have left this out of the manuscript and will leave it for future work.**

*L448 add numbers to compare the load and the content (and possibly a picture 😜)*

**R: The values are all shown on Figure 6. We do not have sediment load values for large chunks of ice collected specifically for visible sediment because these layers are typically a "sediment slurry" and it would be difficult to volume-weight them without collecting immense amounts of ice. Looking at Figure 6, it is clear that the measured sediment loads in ice are extremely low. We do not have a clear photograph of the embedded sediment as it is difficult to capture compared to external sediment smears.**

*L449 necessary for what?*

**R: The purpose of the text was to measure the concentrations of nutrients and trace metals in ice, but Fe and Mn are present at much higher concentrations than other elements and the datasets are thus much larger, we therefore mainly focused on using these elements to look at (micro)nutrient ratios, other elements of course still remain of interest but we have removed them from the text to streamline the structure so this comment is not relevant anymore.**

*L458 add numbers*

*L460 by how much? Discussion?*

**R: We already added the numbers. In Supplementary Table 5 a full list was provided.**

*L466 can or could? Are some ways more common than others?*

**R: As this is a research paper, we think that "could" is correct here.**

*L470 add Ref*

**R: The specific references are given in the next 4 sentences.**

*L473 more sensitive?*

**R: Yes, they are more sensitive to the method used as stated. Phosphate is more challenging to measure due to low concentrations so the values produced by an analyst are more sensitive to the exact protocol used.**

*L478 only Greenland again – all the samples from there? Discuss regions separately? Or by catchment etc, jumping*

**R: We just gave this as an example, it is the better surveyed region for these measurements, now stated 'for example'.**

*L482 how sshown in this figure?*

**R: Figure 6, as per the legend, shows that only dSi had any correlation with sediment load, whereas nitrate and phosphate did not.**

*L483 add Ref*

**R: This was added.**

*L486 interesting! And compared to the floating ones if possibleà use info to discuss*

**R: At the present time, this is not possible. Plus a key problem would be that most large icebergs there ground and unground regularly, so it would require a whole funded program just to attempt some sort of tracking and sampling activity.**

*L506 and L514how to be seen?*

**R: We don't understand this, the tables quoted give the nutrient concentrations — which show some high values for dSi.**

*L519 add Ref*

*L521 add Ref*

**R: This was added.**

*L527 have been*

**R: This was edited.**

*L531 very interesting!*

*Add references to this paragraph*

**R:    We have added general references to support the comments about the origins of Fe, Mn, and sediment.**

*L539 more easily observed- why and how?*

**R: The Arctic is more accessible than the Antarctic for many logistical reasons. There are many populated areas in the Arctic where anyone can walk along the coast and see icebergs, human presence in Antarctica is obviously more limited so if you tally iceberg observations people have made within a few metres of icebergs, they are largely Arctic observations.**

*L544 where are these 3 bays? And 3 Bays but only two numbers…?*

**R: We give a mean and median for the samples for all of the bays collectively for brevity, the values for each area individually are now given as well alongside. We add a note for clarity that these are all WAP sites and now give the 3 averages for the separate sites as well.**

*L553 not sure how to see this in Fig 5*

**R: Regression details are added to Fig. 5 (now Fig. 6) so it shows the formula and $R^2$ for each line fit (those with no significant correlation are not plotted as noted).**

*L572 add Ref*

**R: To our knowledge there are no references to explicitly quantify this, it is just a well-known field observation that icebergs, especially small ones, roll.**

*L577 ice fragments- just from parts discharge or the whole floating ice?*

**R: Generally parts as stated.**

*L577 pieces of ice*

**R: This was edited**

*L578 how rapidly washed out (give numbers and discuss the possible processes*

**R: This is not part of the main story, just an interesting observation. Thus, we do not think it is possible to develop this story further within this text and would rather leave room for other groups to develop this.**

*L587 any evidence? Ref. to be discussed*

**R: To our knowledge this is a new observation based on this work.**

*L588 part of the conclusion rather read like a new or further discussion, not concluding and closing this whole story. Paragraph L 618 feels thrown in and rather discussion as Table 1 needs to be explained for example*

**R: We have moved material from the original conclusion to a general flux summary at the end of the discussion and rewritten a sharper, shorter conclusion.**

---

## Editor Decision (ED1)

**Editor assessment of revised manuscript: Krause et al. Icebergs**

Dear authors,

Thank you for your thorough revision of the manuscript and response to the reviewers' requests. I find the manuscript much improved, but would like to request some further amendments to improve clarity, slightly reduce the length, and ensure it is suitable for the journal's audience.

The requested changes are listed below, and refer to the line numbers in the tracked changes manuscript uploaded. My main criticism is in the use of 'atmospheric origin' as an explanation of NO3 and PO4 sources. I understand that you can argue that these compounds only reach the icebergs' base via incorporation into the ice matrix, either as snow or via cryoconite or supraglacial sediment. However, in the nomenclature of glacial literature it is rather misleading, since atmospheric tends to denote transport from the upper atmosphere to ice/snow surfaces, rather than anything from the air (which could include wind-blown debris). Instead, I request that they are labelled as sourced from ice sheet or glacier surfaces. The origin may then be from chemical or biological scavenging from supraglacial sediment (which may be windblown, and thus technically atmospheric, although not strictly from the atmosphere), or from aerosol deposition (N species), or from precipitation (N species). I give specific suggestions below.

Thank you for your contribution to the Cryosphere, I look forward to reading the next iteration of the manuscript.

Editor, Dr Liz Bagshaw.

**Requested amendments**

L19: add 'low' prior to availability of Fe and Mn

L29-30: rewritten sentence does not make sense 'whilst total dissolvable Fe and Mn retained a strong relationship with sediment load, where weaker relationships were observed…' Please correct (could just remove 'whilst'). Suggest also removing 'retained' – not necessary.

L34: remove 'however' and unclear what you mean by meltwater flux here. Suggest removing from abstract since it is a minor component of your work.

L41 and throughout manuscript: I don't like the use of 'atmospheric origin'. P is not sourced from the atmosphere – I see that you argue that it is from cryoconite thus 'atmosphere' but I think this is too confusing. Instead, can you note that N and P are likely from glacier and ice sheet surfaces, where Fe, Mn and occasionally Si are from en- or subglacial sources.

L61: arguably the polar oceans are the cryosphere, so please change to 'interface between glaciers and ice sheets and the ocean' or further simplify to 'marine-terminating ice'. This whole section could be simplified to 'Icebergs are reported to be sources of fertilizing nutrients to low productivity zones of the ocean, particularly in the southern ocean (Refs). Fe is thought to be the main nutrient limiting phytoplankton growth, so changes to regional Fe supply can have widespread ecosystem impacts. Whilst icebergs are recognized as important sources of Fe (Refs), the sensitivity of this source to climatic impacts (IF YOU ACTUALLY DO THIS?? IF NOT, CUT) and the relative importance of delivery of other critical micro- and macro-nutrients remains to be analysed. Recent work has suggested that low dissolved manganese concentrations….'

L96-109: I dislike the argument that nutrients are atmospheric in origin. Whilst this link can be tenuously proven, I think it can be simplified as 'nutrients in icebergs are either sourced from the ice crystal structure (Fischer) or from sediments either deposited on the ice surface or entrained in the interior or basal structure. Internal cycling may redistribute these nutrients and affect their relative abundances....'

L148: cut the first sentence and ensure these references are incorporated elsewhere if they are critical to your narrative. The paper is too long to include 'commented on' – this is a paper not a PhD thesis.

Figure 1: very nice, thank you for this addition. Can you plot one above the other so we can resolve some of the detail?

L392: I think rather than 'runoff-sediment interaction is limited' you could explicitly state that there is unlikely to be significant subglacial chemical weathering, since this is a cryosphere journal.

L450: this sentence is very awkwardly expressed. Would recommend simplifying: 'the similarity between nutrient ratios in sea ice (Henley et al) and some of our samples suggest seawater is an important contributor to iceberg nutrients, albeit unevenly distributed because of the differing structure of sea ice and glacial ice (refs).' Recommend cutting L453-460.

Figure 5: can you note the distance that defines 'inshore' and 'offshore' in your caption?

L534: once again, I request removing 'atmospheric' origin of P. Suggest just cutting L534-535.

Figure 6a: not quite sure what this is showing. Could remove to make more space for 6b which is arguably more interesting.

L533: remove 'when approaching ice fragments'.

L562: misnumbered figure? Is there where you use 6a?

L569: remove 'atmospheric deposition of NO3 and PO4 varies regionally' and 'reported concentrations of PO4 are more sensitive to the method used due to universally low concentrations' – neither are required for your argument and rather muddy the water. The statement of PO4 concentrations in ice cores from Kjaer et al is sufficient.

L582: I don't think it can be argued that no PO4 can be released from subglacial weathering. It may be that PO4 is taken up prior to measurement, so all remains bound in organic phases. Regardless, I don't think this affects your argument and I suggest just removing L582-584 ('in contrast, no, or very limited release of NO3 or PO4 is expected from weathering, which is supported by the correlations').

L605: suggest adding 'some basal layers are lost prior to...' since not all layers will be scoured

L613: 'glacial origin' rather atmospheric? Or cut this sentence again.

L639: latter not later. Can just stick with 'the former generally having higher sediment loads'

L641: overlong correction here. Keep it simple: 'Arctic icebergs are generally smaller because they are typically sourced from tidewater glacier fronts rather than calved from larger ice shelves. They are also logistically easier to observe and access than Antarctic icebergs.'

L657: misnumbered figure?

L667: I think this is cool, but you've already discussed it so I think this paragraph can be cut

L691: not sure the cryoconite explanation helps here. Suggest just leaving with 'the mechanism of this process remains unclear'.

L746: other N 'phases' rather than 'sources'. I also wonder about adding P to this sentence, since a survey of organic P fascinating. Thus it would become 'considering the universally low concentrations present in icebergs, other phases of N or P (e.g. DON, NH4, DOP) may be important'

L759: 'below or at the standard analytical detection limit for PO4 and NO3' – to make it clear to readers who are just skimming your conclusions which macronutrients you assessed!

---

## Author Response (AR2)

**Editor assessment of revised manuscript: Krause et al. Icebergs**

*Dear authors,*

*Thank you for your thorough revision of the manuscript and response to the reviewers' requests. I find the manuscript much improved, but would like to request some further amendments to improve clarity, slightly reduce the length, and ensure it is suitable for the journal's audience.*

*The requested changes are listed below, and refer to the line numbers in the tracked changes manuscript uploaded. My main criticism is in the use of 'atmospheric origin' as an explanation of NO3 and PO4 sources. I understand that you can argue that these compounds only reach the icebergs' base via incorporation into the ice matrix, either as snow or via cryoconite or supraglacial sediment. However, in the nomenclature of glacial literature it is rather misleading, since atmospheric tends to denote transport from the upper atmosphere to ice/snow surfaces, rather than anything from the air (which could include wind-blown debris). Instead, I request that they are labelled as sourced from ice sheet or glacier surfaces. The origin may then be from chemical or biological scavenging from supraglacial sediment (which may be windblown, and thus technically atmospheric, although not strictly from the atmosphere), or from aerosol deposition (N species), or from precipitation (N species). I give specific suggestions below.*

*Thank you for your contribution to the Cryosphere, I look forward to reading the next iteration of the manuscript.*

*Editor, Dr Liz Bagshaw.*

R: We thank the Editor for their time, consideration and especially for their comments to help prove readability for The Cryosphere's readers. We have adjusted the terminology throughout as requested.

**Requested amendments**

*L19: add 'low' prior to availability of Fe and Mn*
R: Added.

*L29-30: rewritten sentence does not make sense 'whilst total dissolvable Fe and Mn retained a strong relationship with sediment load, where weaker relationships were observed…' Please correct (could just remove 'whilst'). Suggest also removing 'retained' – not necessary.*
R: Rewritten as "Total dissolvable Fe and Mn had a strong relationship with sediment load, whereas…"

*L34: remove 'however' and unclear what you mean by meltwater flux here. Suggest removing from abstract since it is a minor component of your work.*
R: Rewritten as "Dissolved Mn was present at higher dMn:dFe ratios, with fluxes from melting ice…"

*L41 and throughout manuscript: I don't like the use of 'atmospheric origin'. P is not sourced from the atmosphere – I see that you argue that it is from cryoconite thus 'atmosphere' but I think this is too confusing. Instead, can you note that N and P are likely from glacier and ice sheet surfaces, where Fe, Mn and occasionally Si are from en- or subglacial sources.*
R: Rephrased as suggested for clarity. We have used the term as suggested above "the ice matrix" in contrast to additions associated with englacial sediment. Based on the editor's suggestion, we have edited the text to: "Our results suggest that $NO_x^-$ and $PO_4^{3-}$ concentrations measured in calved icebergs originate from glacier and ice sheet surface processes which determined the ice matrix . Conversely, high Fe and Mn, and occasionally high dSi concentrations, are associated with englacial sediment, which experiences limited biogeochemical processing prior to release into the ocean."

*L61: arguably the polar oceans are the cryosphere, so please change to 'interface between glaciers and ice sheets and the ocean' or further simplify to 'marine-terminating ice'. This whole section could be*

*simplified to 'Icebergs are reported to be sources of fertilizing nutrients to low productivity zones of the ocean, particularly in the southern ocean (Refs). Fe is thought to be the main nutrient limiting phytoplankton growth, so changes to regional Fe supply can have widespread ecosystem impacts. Whilst icebergs are recognized as important sources of Fe (Refs), the sensitivity of this source to climatic impacts (IF YOU ACTUALLY DO THIS?? IF NOT, CUT) and the relative importance of delivery of other critical micro- and macro-nutrients remains to be analysed. Recent work has suggested that low dissolved manganese concentrations….'*

R: Rephrased as "At the interface between the marine-terminating ice and ocean". We agree the revised Introduction can be shortened for brevity and have done so following the Editor's helpful suggestion following the above structure suggestion to trim parts of the text.

*L96-109: I dislike the argument that nutrients are atmospheric in origin. Whilst this link can be tenuously proven, I think it can be simplified as 'nutrients in icebergs are either sourced from the ice crystal structure (Fischer) or from sediments either deposited on the ice surface or entrained in the interior or basal structure. Internal cycling may redistribute these nutrients and affect their relative abundances….'*
R: As above, we agree that this would be confusing to The Cryosphere readership and have edited throughout referring instead to incorporation into the ice matrix structure.

*L148: cut the first sentence and ensure these references are incorporated elsewhere if they are critical to your narrative. The paper is too long to include 'commented on' – this is a paper not a PhD thesis.*
R: As suggested, we have trimmed the introduction using the suggested structure by the editor to make the text of the first part shorter and more to the point (changes are tracked, the Introduction is now 1/5 shorter than the R2 version).

*Figure 1: very nice, thank you for this addition. Can you plot one above the other so we can resolve some of the detail?*
R: Figure orientation changed as suggested.

*L392: I think rather than 'runoff-sediment interaction is limited' you could explicitly state that there is unlikely to be significant subglacial chemical weathering, since this is a cryosphere journal.*
R: Edited to read: "such that subglacial chemical weathering is probably limited".

*L450: this sentence is very awkwardly expressed. Would recommend simplifying: 'the similarity between nutrient ratios in sea ice (Henley et al) and some of our samples suggest seawater is an important contributor to iceberg nutrients, albeit unevenly distributed because of the differing structure of sea ice and glacial ice (refs).'*
R: We have rephrased the sentence, as this is not what we intended to communicate. The nutrient ratios and concentrations in these cases suggest a seawater source, but the salinity data does not. Sentence re-written as: "The ratio of $NO_x^-$: $PO_4^{3-}$:dSi in sea ice is strong evidence that nutrients in sea ice have a primarily saline origin (Henley et al., 2023)."

*Recommend cutting L453-460.*
R: We are aware of some new (unpublished) work from Greenland which suggests marine ice formation is much more common than presently thought (it is not presently thought to be formed anywhere in south Greenland). If it was formed in some Greenland fjords over winter, it might would explain the above issue with nutrients, so we prefer to keep sentence 453, but we have cut 455–460 as suggested.

*Figure 5: can you note the distance that defines 'inshore' and 'offshore' in your caption?*
R: Edited as suggested: "Inshore samples within 1 km of the coastline, whereas offshore values were all >15 km away from the coastline". For consistency we added the same detail to the methods section.

*L534: once again, I request removing 'atmospheric' origin of P. Suggest just cutting L534-535.*
R: Sentence removed as suggested.

*Figure 6a: not quite sure what this is showing. Could remove to make more space for 6b which is arguably more interesting.*
R: We have modified the figure to include just 6b as requested.

*L533: remove 'when approaching ice fragments'.*

R: Edited as requested.

*L562: misnumbered figure? Is there where you use 6a?*
R: Correct, as this is no longer in the Figure 6, now edited to read: "It should be noted that randomly collected samples had much lower sediment loads"

*L569: remove 'atmospheric deposition of NO3 and PO4 varies regionally' and 'reported concentrations of PO4 are more sensitive to the method used due to universally low concentrations' – neither are required for your argument and rather muddy the water. The statement of PO4 concentrations in ice cores from Kjaer et al is sufficient.*
R: Lines 569–572 Deleted as suggested.

*L582: I don't think it can be argued that no PO4 can be released from subglacial weathering. It may be that PO4 is taken up prior to measurement, so all remains bound in organic phases. Regardless, I don't think this affects your argument and I suggest just removing L582-584 ('in contrast, no, or very limited release of NO3 or PO4 is expected from weathering, which is supported by the correlations').*
R: We agree with this and for brevity have deleted 582–584 as suggested.

*L605: suggest adding 'some basal layers are lost prior to…' since not all layers will be scoured*
R: Edited to: "that basal layers are largely lost prior to, or rapidly following, iceberg calving"

*L613: 'glacial origin' rather atmospheric? Or cut this sentence again.*
R: We now use the phrase suggested by the Editor, "ice matrix" .

*L639: latter not later. Can just stick with 'the former generally having higher sediment loads'*
R: Edited to read: "with the former generally having higher sediment loads."

*L641: overlong correction here. Keep it simple: 'Arctic icebergs are generally smaller because they are typically sourced from tidewater glacier fronts rather than calved from larger ice shelves. They are also logistically easier to observe and access than Antarctic icebergs.'*
R: Edited as suggested.

*L657: misnumbered figure?*
R: We have corrected this to reference Fig. 6.

L667: I think this is cool, but you've already discussed it so I think this paragraph can be cut
R: Removed as suggested.

L691: not sure the cryoconite explanation helps here. Suggest just leaving with 'the mechanism of this process remains unclear'.
R: Edited to read: "The mechanism of this process is unclear."

L746: other N 'phases' rather than 'sources'. I also wonder about adding P to this sentence, since a survey of organic P fascinating. Thus it would become 'considering the universally low concentrations present in icebergs, other phases of N or P (e.g. DON, NH4, DOP) may be important'
R: Yes, this is correct, edited to read: "$NO_x^-$ and $PO_4^{3-}$ as sources of bioaccessible nitrogen and phosphorous, but considering the universally low concentrations present in icebergs, other N and P sources (e.g. DON- Dissolved Organic Nitrogen, DOP- Dissolved Organic Phosphorous, and $NH_4$) may be relatively important (Parker et al., 1978)." We think "sources" is a correct term as we are phrasing "sources of N and P to biota".

L759: 'below or at the standard analytical detection limit for PO4 and NO3' – to make it clear to readers who are just skimming your conclusions which macronutrients you assessed

R: Edited for clarity to read: "below the standard analytical detection limit- especially for $PO_4^{3-}$ and dSi"